# Two mechanisms repress cyclin B1 translation to maintain prophase arrest in mouse oocytes

Shiya Cheng [1,3] & Melina Schuh [1,2] ✉

In mammals, oocytes are arrested in prophase of meiosis I for long periods of time. Prophase arrest is critical for reproduction because it allows oocytes to grow to their full size to support meiotic maturation and embryonic development. Prophase arrest requires the inhibitory phosphorylation of the mitotic kinase CDK1. Whether prophase arrest is also regulated at the translational level is unknown. Here, we show that prophase arrest is regulated by translational control of dormant cyclin B1 mRNAs. Using Trim-Away, we identify two mechanisms that maintain cyclin B1 dormancy and thus prophase arrest. First, a complex of the RNA-binding proteins DDX6, LSM14B and CPEB1 directly represses cyclin B1 translation through interacting with its 3'UTR. Second, cytoplasmic poly(A)-binding proteins (PABPCs) indirectly repress the translation of cyclin B1 and other poly(A)-tail-less or short-tailed mRNAs by sequestering the translation machinery on long-tailed mRNAs. Together, we demonstrate how RNA-binding proteins coordinately regulate prophase arrest, and reveal an unexpected role for PABPCs in controlling mRNA dormancy.

Women are born with a finite pool of oocytes, which are stored in the ovary and constitute the ovarian reserve. During storage, the oocytes are arrested in the prophase stage of meiosis I. Prophase arrest can last for decades in humans and months in mice. The prolonged prophase arrest also allows the oocyte sufficient time to grow in size and accumulate maternal components for progression through the meiotic divisions and early embryonic development[1]. Oocytes that resume meiosis prematurely are unable to complete meiotic maturation and early embryonic development properly[2,3]. Therefore, prophase arrest is crucial for successful human reproduction.

Prophase arrest is known to be regulated at the post-translational level[4–6]. A G protein-coupled receptor-G protein-adenylyl cyclase (GPR-Gs-ADCY) cascade generates cAMP in the oocyte. Granulosa cell-derived cGMP diffuses into the oocyte, where it inhibits the cAMP phosphodiesterase (cAMP-PDE) and thus prevents cAMP hydrolysis. These two

effects work together to maintain a relatively high level of cAMP in the oocyte, which activates PKA. PKA further activates the kinases Wee1B and Myt1 and inactivates the phosphatase CDC25B, which in turn promotes the inhibitory phosphorylation of CDK1 at Thr14 and Tyr15, thereby arresting the oocyte in prophase. However, whether prophase arrest is also regulated at the translational level remains unknown.

As the oocyte grows, it also accumulates a large number of maternal mRNAs. These maternal mRNAs are essential for oocyte meiotic maturation and early embryonic development, as full-grown oocytes are transcriptionally silent and can only use the stored mRNAs to synthesize new proteins[1,7]. RNA storage compartments, including P-body-like granules and the mitochondria-associated ribonucleoprotein domain (MARDO), have been identified at different stages of oocyte growth[8,9]. The majority of the stored mRNAs are dormant during oocyte growth and are only activated when the oocyte resumes

[1]Department of Meiosis, Max Planck Institute for Multidisciplinary Sciences, 37077 Göttingen, Germany. [2]Cluster of Excellence "Multiscale Bioimaging: from Molecular Machines to Networks of Excitable Cells" (MBExC), University of Göttingen, 37077 Göttingen, Germany. [3]Present address: Hubei Provincial Key Laboratory of Developmentally Originated Disease, TaiKang Center for Life and Medical Sciences, School of Basic Medical Sciences, Wuhan University, 430072 Wuhan, China. ✉e-mail: melina.schuh@mpinat.mpg.de

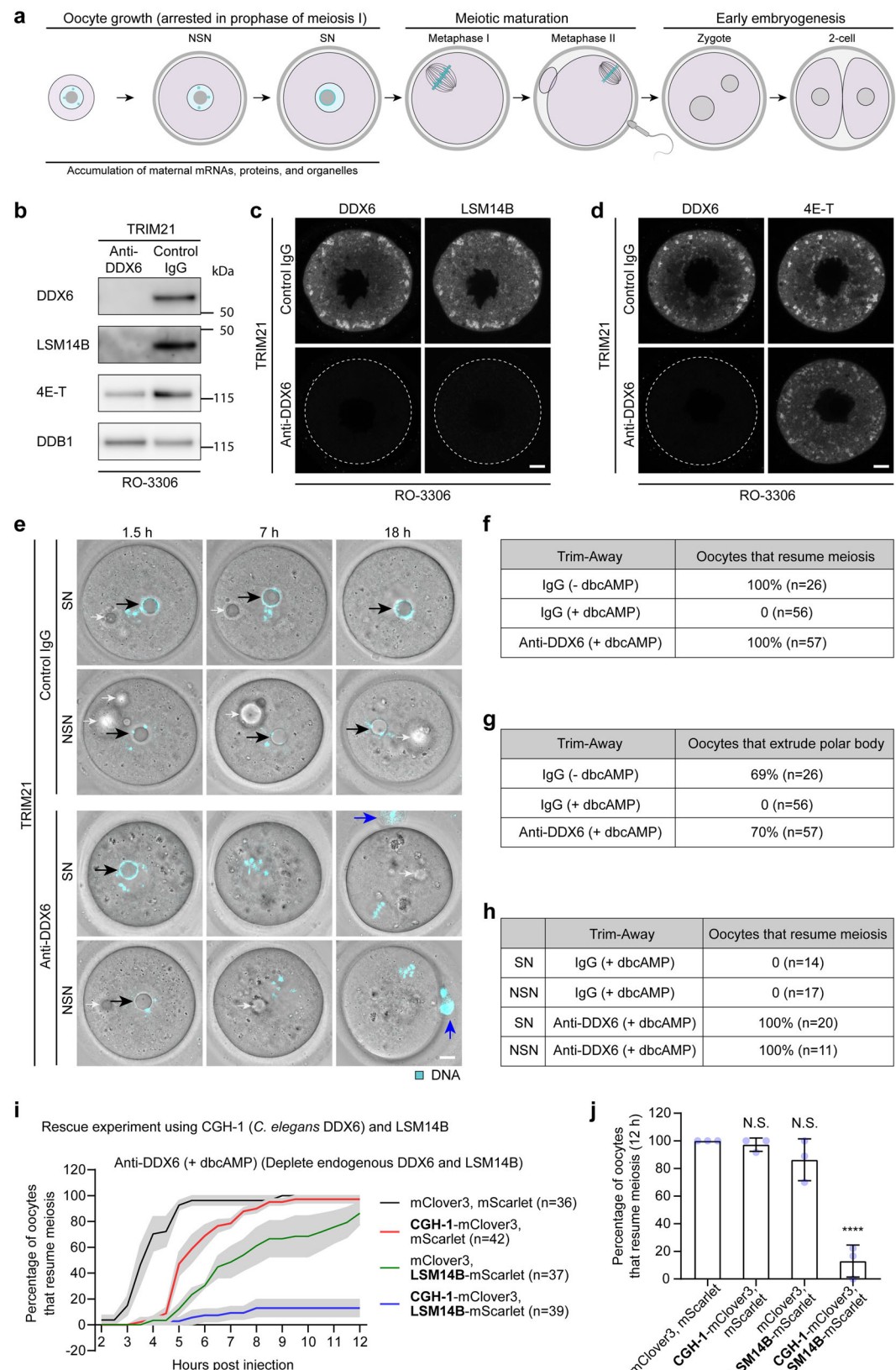

meiosis or after fertilization[7]. RNA-binding proteins play a central role in the translational control of the stored mRNAs and are therefore the most promising candidates for controlling prophase arrest at the translational level.

Through gene knockout, a number of RNA-binding proteins required for oocyte or embryo development have been identified in mice, such as PABPC1L[10], YBX2[11], ZAR1[9,12,13], ZAR2[12], YTHDF2[14], IGF2BP2[15], METTL3[16], ELAVL2[17], PATL2[18,19], DDX6[20], LSM14B[21–24], 4E-T[25], AND EIF4E1B[26,27]. These studies have greatly advanced our understanding of maternal mRNA storage and translation. One feature of gene knockout is that it may affect oocyte development from a very early stage, and thus the phenotypes observed in full-grown oocytes

**Fig. 1 | Acute depletion of DDX6 or LSM14B causes premature resumption of meiosis. a** Schematic representation of mouse oocyte growth, meiotic maturation, and early embryonic development. **b** Western blot analyses of mouse oocytes showing co-depletion of DDX6 and LSM14B upon DDX6 Trim-Away using a DDX6 monoclonal antibody. Oocytes were incubated for 6 h in the presence of 10 μM RO-3306 after injection of *Trim21* mRNA and DDX6 monoclonal antibody. **c**, **d** Representative immunofluorescence images of mouse oocytes showing co-depletion of DDX6 and LSM14B upon DDX6 Trim-Away. Oocytes were incubated for 6 h in the presence of 10 μM RO-3306 after injection of *Trim21* mRNA and DDX6 monoclonal antibody. The dashed line demarcates the oocyte. **e** Representative stills from time-lapse movies of control and DDX6/LSM14B-depleted mouse oocytes stained with 5-SiR-Hoechst. Oocytes were incubated with dbcAMP throughout the experiment. Black arrows point to nucleolus of prophase-arrested oocytes. Blue arrows point to polar bodies of mature eggs. Smaller white arrows point to injection oil droplets that are out of focus or in focus. Cyan, DNA. **f** Percentage of control and DDX6/LSM14B-depleted mouse oocytes that resume

meiosis in 18 h in the presence or absence of dbcAMP. **g** Percentage of control and DDX6/LSM14B-depleted mouse oocytes that extrude polar bodies in 18 h in the presence or absence of dbcAMP. **h** Percentage of control and DDX6/LSM14B-depleted SN or NSN oocytes that resume meiosis in 18 h in the presence of dbcAMP. **i** Percentage of DDX6/LSM14B-depleted mouse oocytes that resume meiosis in the presence of dbcAMP. Endogenous DDX6 and LSM14B were acutely depleted by Trim-Away. Exogenous mClover3, mScarlet, mClover3-CGH-1, and LSM14B-mScarlet were expressed in oocytes in different combinations. **j** Percentage of DDX6/LSM14B-depleted mouse oocytes that resume meiosis in 12 h in the presence of dbcAMP. Data are from two (**b–e**, **h**), four (**f**, **g**), or three (**i**, **j**) independent experiments. "*n*" indicates the number of analyzed oocytes. Data are shown as mean ± SEM (**i**) or mean ± SD (**j**). *P*-values were calculated using one-way ANOVA with Tukey's *post-hoc* test. *P*-values are designated as ****$P < 0.0001$. Nonsignificant values are indicated as N.S. Scale bars, 10 μm. Source data are provided as a Source Data file.

represent cumulative effects over a long period. Additionally, during the extended growth phase, which spans about 3 weeks in mice, knockout oocytes may develop compensatory mechanisms to overcome the loss of a gene[28]. Biochemical studies in *Xenopus* suggest the presence of a translational repression complex containing CPEB1, Xp54 (mouse DDX6), Pat1, RAP55 (mouse LSM14A or LSM14B), and 4E-T in prophase-arrested oocytes[29,30]. A similar complex may also exist in mouse oocytes[21]. However, it remains unclear whether this complex is related to prophase arrest due to the technical characteristics described above.

Trim-Away[31], an antibody-based targeted protein degradation technology, can complement gene knockout. With Trim-Away, a protein can be depleted from full-grown oocytes within a few hours[31], eliminating concerns about effects on early-stage oocytes or the development of potential compensatory mechanisms. Moreover, prophase-arrested oocytes lack mechanisms for rapid mRNA degradation[7,32] and therefore their mRNA levels are unlikely to be significantly changed in such a short time, which facilitates the analysis of the effects on mRNA translation. In this study, by combining Trim-Away-based loss-of-function analyses and imaging-based assays, we demonstrate how prophase arrest is regulated at the translational level and also reveal an unexpected mechanism by which poly(A)-binding proteins regulate translation of mRNAs with different poly(A) tail lengths.

## Results

### DDX6 and LSM14B are essential for maintaining prophase arrest

DDX6 is highly expressed in mouse oocytes, but its function is not fully understood. To investigate the direct effects of DDX6 on oocyte prophase arrest and meiotic maturation, we used Trim-Away to acutely deplete DDX6 in late-stage mouse oocytes (Fig. 1a). Western blot and immunofluorescence staining showed that Trim-Away of DDX6 using a DDX6 monoclonal antibody depleted both DDX6 and LSM14B to almost undetectable levels, while having little effect on 4E-T[33], which interacts with both DDX6 and LSM14B (Fig. 1b–d and Supplementary Fig. 1a, b). DDX6 Trim-Away also had no obvious effect on YBX2 and ZAR1 (Supplementary Fig. 1c, d). Similar results were observed when we performed the experiments with a polyclonal DDX6 antibody (Supplementary Fig. 1e). Thus, LSM14B proteins were associated with DDX6 and co-depleted with DDX6 after DDX6 Trim-Away in mouse oocytes.

Oocytes are arrested in prophase until they reach full size and accumulate sufficient maternal components (Fig. 1a)[1,9,34]. Prophase arrest relies on the PKA signaling cascade. Therefore, supplementation of the PKA activator dibutyryl cyclic AMP (dbcAMP) maintains isolated oocytes in prophase. Strikingly, acute depletion of DDX6 and LSM14B resulted in spontaneous resumption of meiosis in the presence of dbcAMP (Fig. 1e, f). These oocytes extruded polar bodies at a similar

rate to control oocytes cultured in dbcAMP-free medium (Fig. 1e, g). Thus, DDX6 and LSM14B are essential for the maintenance of prophase arrest in mouse oocytes. Full-grown oocytes are called SN (surrounded nucleolus) oocytes because their DNA is highly condensed around the nucleolus, forming ring-like structures, while oocytes that are not fully grown are called NSN (non-surrounded nucleolus) oocytes because their DNA is less condensed and does not cluster around the nucleolus (Fig. 1a). Only full-grown oocytes are capable of normal meiotic divisions and early embryonic development[2,3]. We found that acute depletion of DDX6 and LSM14B also caused a spontaneous resumption of meiosis in NSN oocytes (Fig. 1e, h), suggesting that DDX6 and LSM14B prevent the premature resumption of meiosis during oocyte growth.

DDX6 Trim-Away depleted both DDX6 and LSM14B (Fig. 1b–d and Supplementary Fig. 1a, b). To test whether both proteins are required to maintain prophase arrest, we performed rescue experiments. Since exogenously expressed mouse DDX6 was also recognized by the DDX6 antibody and cleared upon Trim-Away, we used CGH-1, the *C. elegans* DDX6, which is highly similar to mouse DDX6 but not recognized by the DDX6 antibody, for the rescue experiments (Supplementary Fig. 1f, g). Oocytes depleted of endogenous DDX6 and LSM14B could only restore prophase arrest when CGH-1 and LSM14B were co-expressed (Fig. 1i, j). By contrast, expression of CGH-1 or LSM14B alone did not restore prophase arrest (Fig. 1i, j). Thus, both DDX6 and LSM14B are required to maintain prophase arrest and prevent premature resumption of meiosis during oocyte development.

### DDX6 and LSM14B repress cyclin B1 translation to maintain prophase arrest

To understand how DDX6 and LSM14B regulate oocyte meiotic progression, we focused on the expression of several key cell cycle regulators, cyclin B1, MOS and CDK1. Oocytes accumulate large amounts of cyclin B1 and *Mos* mRNAs during growth. These mRNAs are translationally repressed in prophase-arrested oocytes and are only activated when oocytes resume meiosis[35,36]. We found that acute depletion of DDX6 and LSM14B caused premature accumulation of cyclin B1 proteins within 6 h of injection of *Trim21* mRNA and DDX6 antibody, whereas it had little effect on MOS and CDK1 protein levels during this period when oocytes were maintained in prophase by supplementing the CDK1 inhibitor RO-3306 (Fig. 2a, b). CDK1 is phosphorylated at Thr14 and Tyr15 in prophase and dephosphorylated when oocytes resume meiosis (Supplementary Fig. 2a)[4–6]. Acute depletion of DDX6 and LSM14B did not affect CDK1 phosphorylation when oocytes were maintained in prophase in the presence of RO-3306 (Fig. 2c). Thus, the premature accumulation of cyclin B1 protein is the earliest event observed upon acute depletion of DDX6 and LSM14B.

To test directly whether DDX6 and LSM14B repress cyclin B1 translation, we modified the recently developed RIBOmap assay[37] to

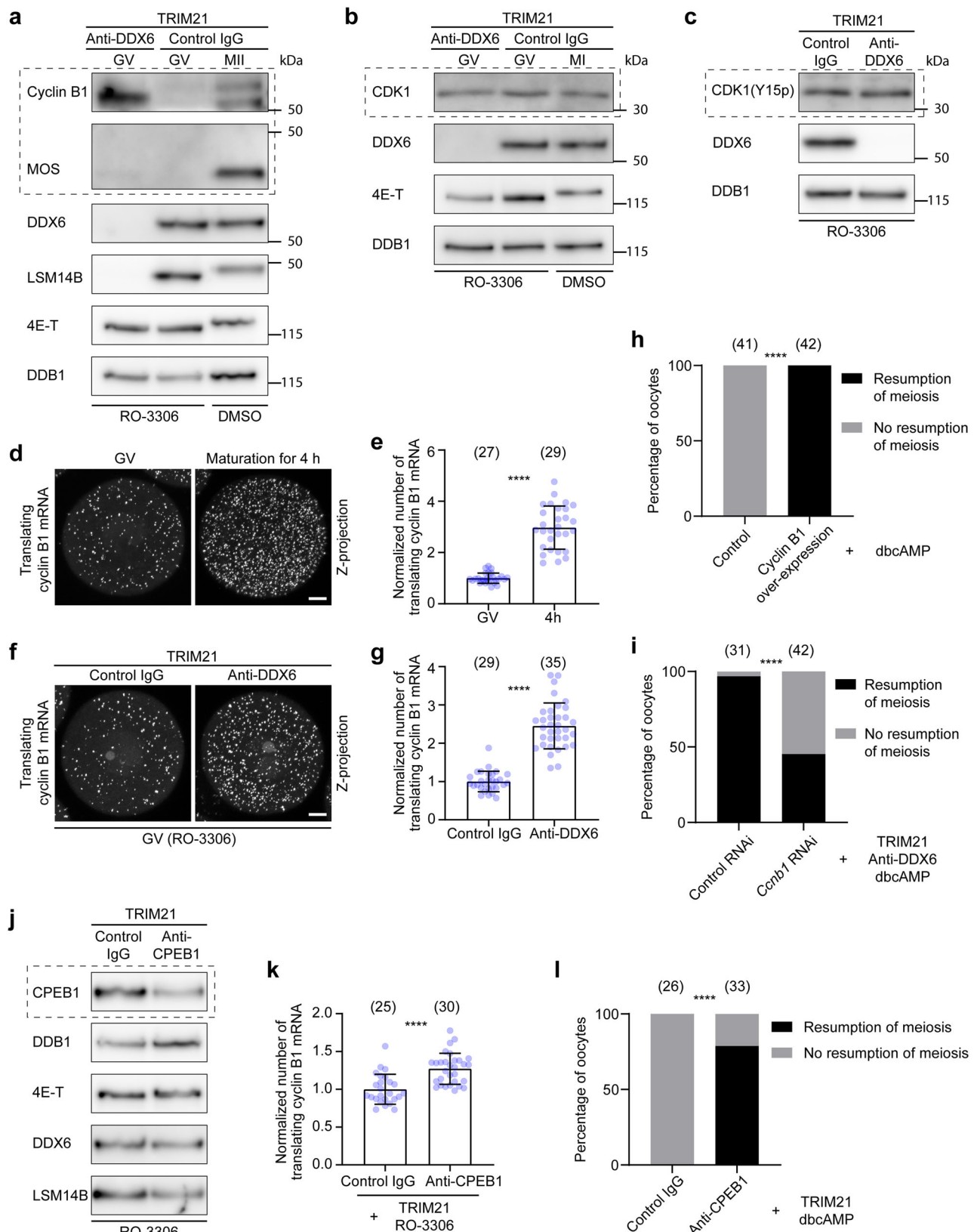

detect translating cyclin B1 mRNAs in mouse oocytes. Translating cyclin B1 mRNAs increased when control oocytes resumed meiosis (Fig. 2d, e), suggesting that the assay works well in mouse oocytes. Acute depletion of DDX6 and LSM14B significantly increased translating cyclin B1 mRNAs in prophase-arrested oocytes (Fig. 2f, g), without affecting total cyclin B1 mRNA levels (Supplementary Fig. 2b).

Thus, DDX6 and LSM14B indeed repress the translation of the stored cyclin B1 mRNAs.

Next, we performed a series of experiments to test whether the premature translation of the stored cyclin B1 mRNAs is the cause of the premature meiotic resumption upon acute depletion of DDX6 and LSM14B. First, expression of exogenous cyclin B1 from mRNA

**Fig. 2 | Acute depletion of DDX6 or LSM14B causes premature translation of cyclin B1, which leads to premature resumption of meiosis. a, b** Western blot analyses of mouse oocytes showing premature accumulation of cyclin B1 upon DDX6 Trim-Away. Oocytes were either incubated for 6 h in the presence of 10 μM RO-3306 to maintain them at the GV stage or incubated for 6 h or 15 h without RO-3306 to mature to the MI or MII stage after injection of *Trim21* mRNA and DDX6 monoclonal antibody. **c** Western blot analyses of mouse oocytes showing that CDK1 phosphorylation at Tyr15 was not affected by DDX6 Trim-Away. Oocytes were incubated for 6 h in the presence of 10 μM RO-3306 after injection of *Trim21* mRNA and DDX6 monoclonal antibody. **d** Representative RIBOmap images of translating cyclin B1 mRNAs in GV and maturing oocytes. **e** Quantification and normalization of the number of translating cyclin B1 mRNAs in (**d**). **f** Representative RIBOmap images of translating cyclin B1 mRNAs in control and DDX6/LSM14B-depleted oocytes. Oocytes were incubated for 6 h in the presence of 10 μM RO-3306 after injection of *Trim21* mRNA and DDX6 monoclonal antibody. **g** Quantification and normalization of the number of translating cyclin B1 mRNAs in (**f**). **h** Percentage of control and cyclin B1-overexpressing mouse oocytes that resume meiosis in 12 h in the presence of dbcAMP. **i** Percentage of DDX6/LSM14B-depleted mouse oocytes that resume meiosis in 12 h in the presence of dbcAMP. Control or *Ccnb1* RNAi was performed before Trim-Away. **j** Western blot analyses of mouse oocytes showing partial depletion of CPEB1 upon CPEB1 Trim-Away. 4E-T, DDX6, and LSM14B were not obviously affected. **k** RIBOmap assay showing the number of translating cyclin B1 mRNAs in control and CPEB1-depleted oocytes. The data were normalized by dividing the values of each group by the mean of the control IgG group. Oocytes were incubated for 6 h in the presence of 10 μM RO-3306 after injection of *Trim21* mRNA and CPEB1 antibody. **l** Percentage of control and CPEB1-depleted mouse oocytes that resume meiosis in 18 h in the presence of dbcAMP. Data are all from two independent experiments. The number of analyzed oocytes is specified in italics. Data are shown as mean ± SD. *P*-values were calculated using unpaired two-tailed Student's *t* test (**e, g, k**) or two-tailed Fisher's exact test (**h, i, l**). *P*-values are designated as ****$P < 0.0001$. Scale bars, 10 μm. Source data are provided as a Source Data file.

promoted meiotic resumption in the presence of dbcAMP (Fig. 2h), mimicking the phenotype observed upon DDX6/LSM14B depletion and aligning with previous findings from cyclin B1 protein injection experiments[38]. Second, RNA interference (RNAi)-mediated knockdown of cyclin B1 mRNA partially suppressed the meiotic resumption caused by DDX6/LSM14B depletion (Fig. 2i and Supplementary Fig. 2c). Moreover, the resumption of meiosis was delayed when oocytes depleted of DDX6 and LSM14B were treated with the CDK1 inhibitor RO-3306 (Supplementary Fig. 2d), which suggests that DDX6 and LSM14B regulate meiotic progression through the cyclin B1-CDK1 complex. RO-3306 was therefore used to prolong prophase in Trim-Away experiments.

CPEB1 is in a complex with DDX6 and LSM14B and is known to repress cyclin B1 translation by interacting with the cytoplasmic polyadenylation elements (CPE) in its 3′UTR[21,29,39]. Knockdown of CPEB1 by Trim-Away also resulted in increased cyclin B1 translation and premature meiotic resumption (Fig. 2j–l), suggesting that CPEB1 may be an adaptor protein that mediates the interaction between DDX6/LSM14B and cyclin B1 mRNA. In addition, acute depletion of DDX6 and LSM14B also increased the translation of several other validated CPEB1-bound mRNAs containing CPEs[35], while having no obvious effect on mRNAs without CPEs (Supplementary Fig. 2e, f).

Together, our data establish that DDX6, LSM14B and CPEB1 repress the translation of the stored cyclin B1 mRNAs to prevent premature resumption of meiosis. This mechanism of translational repression may also apply to other CPEB1-bound mRNAs.

## Acute depletion of PABPCs further enhances cyclin B1 translation

In wild-type oocytes, cyclin B1 is only actively translated when its poly(A) tail is elongated after meiotic resumption[35,36]. We therefore asked whether the premature translation of cyclin B1 upon acute depletion of DDX6 and LSM14B was dependent on poly(A) tail elongation. To address this, we examined the length of the poly(A) tail of cyclin B1 mRNA in oocytes at different stages of meiosis. The poly(A) tail of cyclin B1 was short in prophase-arrested oocytes and was significantly elongated in metaphase I oocytes (Fig. 3a, b). Interestingly, we observed an increase in cyclin B1 translation within 6 hours of injection of *Trim21* mRNA and DDX6 antibody (Fig. 2a, f, g), whereas the length of the poly(A) tail remained unchanged during this period (Fig. 3b). These results show that the premature translation of cyclin B1 mRNA upon acute depletion of DDX6 and LSM14B was independent of poly(A) tail elongation.

Poly(A) tails are known to promote translation[35,40,41]. Although the premature translation of cyclin B1 was independent of poly(A) tail elongation, we hypothesized that the pre-existing short poly(A) tail of cyclin B1 might be important for its increased translation upon acute depletion of DDX6 and LSM14B. To test this hypothesis, we constructed a reporter mRNA by fusing *mClover3* and the 3′UTR of cyclin

B1 to mimic cyclin B1 expression, and injected this reporter mRNA together with *mScarlet* mRNA as injection control (Fig. 3c). Translation of the reporter mRNA was assessed by the signal ratio of mClover3 to mScarlet. When the reporter mRNA contained a short poly(A) tail, acute depletion of DDX6 and LSM14B increased its translation without affecting its mRNA level (Fig. 3d, f and Supplementary Fig. 3a), consistent with the increased translation of endogenous cyclin B1 (Fig. 2a, f, g). However, when the reporter mRNA was tailless, its translation was not altered (Fig. 3e, g). Therefore, the premature translation of cyclin B1 upon acute depletion of DDX6 and LSM14B was indeed dependent on the pre-existing short poly(A) tail.

The expression level of the reporter mRNA was higher when it contained the short poly(A) tail (Fig. 3h), confirming that poly(A) tails promote translation. Poly(A) tails promote translation by recruiting cytoplasmic poly(A)-binding proteins (PABPCs)[40]. We therefore asked whether the premature translation of cyclin B1 upon acute depletion of DDX6 and LSM14B was dependent on PABPCs. Three PABPCs, PABPC1L, PABPC1 and PABPC4 are highly expressed in mouse oocytes[42,43]. Using Trim-Away, we successfully depleted PABPC1L and PABPC4 (Fig. 3i, j). As depletion of PABPCs may affect the translation of any mRNA containing a poly(A) tail, we avoided using the ratio of mClover3 to mScarlet to assess translation when depleting PABPCs. Instead, equal amounts of the reporter mRNA *mClover3-Ccnb1 3′UTR-A₃₄* were injected into all oocytes and the fluorescence intensity was directly used to indicate translation (Fig. 3k). Surprisingly, acute depletion of PABPCs did not suppress the increased translation of cyclin B1 in oocytes depleted of DDX6 and LSM14B (Fig. 3l, m). On the contrary, it further increased its translation without affecting its mRNA level (Fig. 3l, m and Supplementary Fig. 3b). These unexpected data suggest that there is a previously unknown mechanism by which PABPCs regulate translation.

## Acute depletion of PABPCs causes a shift of translation from long-tailed mRNAs to tailless and short-tailed mRNAs

To uncover the unknown mechanism by which PABPCs regulate translation, we examined the translation of mRNAs with different poly(A) tail lengths (Fig. 4a). In the presence of PABPCs, longer poly(A) tails increased translation (Fig. 4b–d and Supplementary Fig. 4a, b). The longer the poly(A) tail, the higher the translation. Strikingly, when PABPCs were acutely depleted, the translation of long-tailed mRNAs decreased, whereas the translation of tailless or short-tailed mRNAs increased (Fig. 4b–d and Supplementary Fig. 4a, b). The sum of translation of long-tailed, short-tailed, and tailless mRNAs was similar in the presence or absence of PABPCs (Fig. 4b–d and Supplementary Fig. 4a, b), suggesting that the total translation capacity of an oocyte is limited. Overall, acute depletion of PABPCs caused a shift of translation from long-tailed mRNAs to tailless and short-tailed mRNAs (Fig. 4b–d and Supplementary Fig. 4a, b). Therefore, acute removal of PABPCs

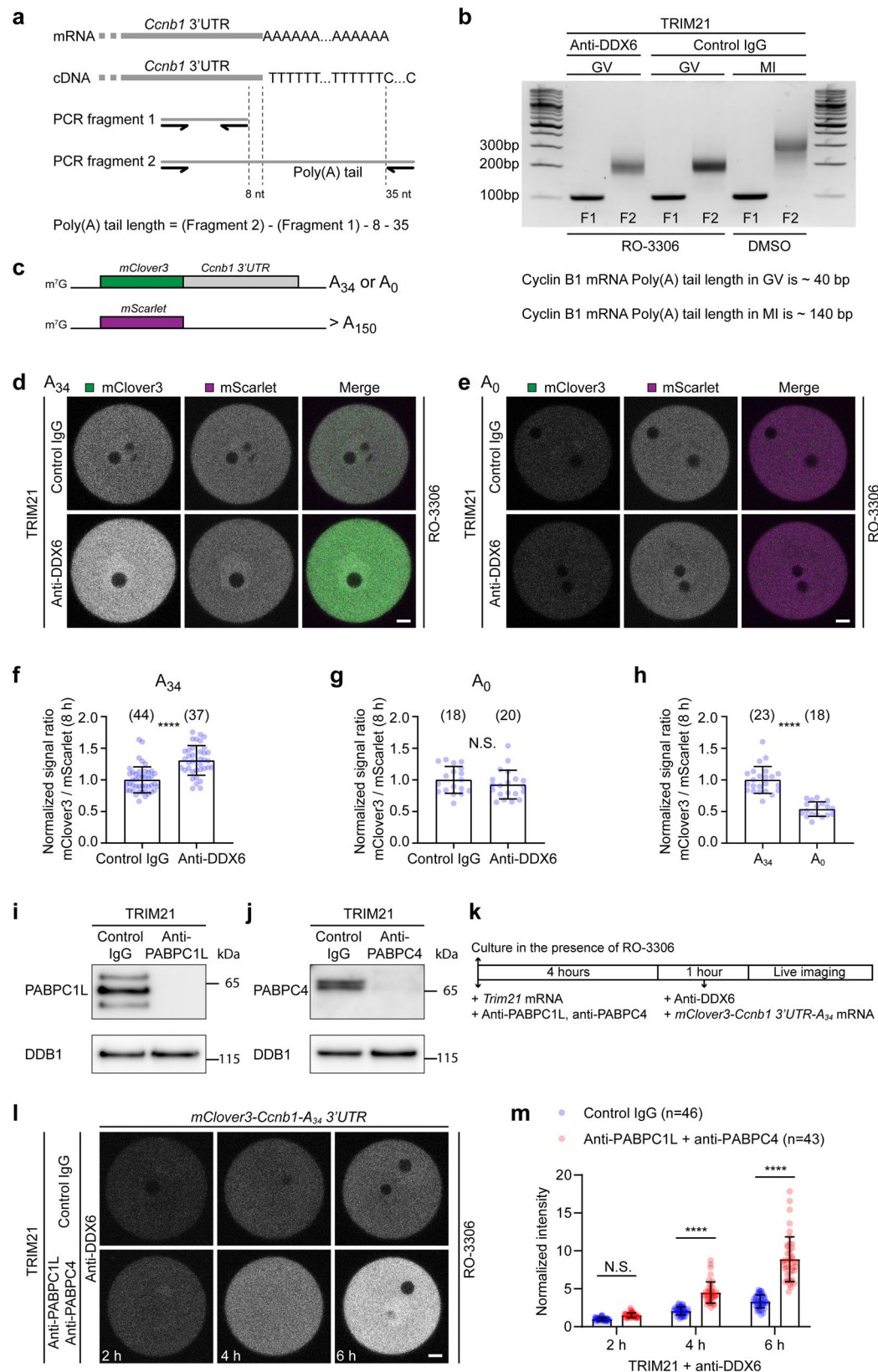

affected both the active translation of long-tailed mRNAs and the dormancy of tailless and short-tailed mRNAs.

To further test how acute depletion of PABPCs affects the translation of endogenous mRNAs, we unbiasedly selected a few published mRNAs with different poly(A) tail lengths based on some reasonable criteria (Fig. 4e and Supplementary Fig. 4c)[35,41] and examined their

translation by performing the modified RIBOmap assay. The translation of dormant endogenous mRNAs with short poly(A) tails, such as *Orc6*, *Btg4* and *Tcl1b4*, was increased upon acute depletion of PABPC1L and PABPC4 (Fig. 4f, g). By contrast, the translation of endogenous mRNAs with long poly(A) tails, such as *Sec61g* and *Tbca*, was decreased (Fig. 4f, i). Interestingly, the translation of *Actb*, which has a poly(A) tail

**Fig. 3 | Acute depletion of PABPCs further enhances the phenotype of acute DDX6/LSM14B depletion. a** Strategy for measuring cyclin B1 mRNA poly(A) tail length. **b** Measurement of the poly(A) tail length of cyclin B1 mRNA in control and DDX6/LSM14B-depleted mouse oocytes. Oocytes were either incubated for 6 h in the presence of 10 μM RO-3306 to maintain them at the GV stage or incubated for 6 h without RO-3306 to mature to the MI stage after injection of *Trim21* mRNA and DDX6 monoclonal antibody. **c** Schematic representation of the *mClove3-Ccnb1 3'UTR* reporter mRNA with or without a short poly(A) tail and the *mScarlet* control mRNA. Translation of the reporter mRNA can be assessed by the signal ratio of mClover3 to mScarlet. **d, e** Representative fluorescence images of control and DDX6/LSM14B-depleted mouse oocytes expressing *mClove3-Ccnb1 3'UTR-A$_{34}$* (**d**) or *mClove3-Ccnb1 3'UTR-A$_{0}$* (**e**) together with *mScarlet* at 8 h post injection. Green, mClover3; magenta, mScarlet. **f, g** Quantification and normalization of the fluorescence intensity ratio of mClover3 to mScarlet in (**d, e**). **h** Quantification and normalization of the fluorescence intensity ratio of mClover3 to mScarlet in control oocytes expressing *mClove3-Ccnb1 3'UTR-A$_{34}$* or *mClove3-Ccnb1 3'UTR-A$_{0}$* together

with *mScarlet* at 8 h post injection. **i, j** Western blot analyses of mouse oocytes showing depletion of PABPC1L (**i**) or PABPC4 (**j**) upon PABPC1L or PABPC4 Trim-Away. Oocytes were incubated for 6 h in the presence of 10 μM RO-3306 after injection of *Trim21* mRNA and antibodies. **k** Schematic diagram of the experiment in (**l**). Acute depletion of PABPC1L and PABPC4 was performed, followed by acute depletion of DDX6 and LSM14B and expression of the reporter mRNA *mClove3-Ccnb1 3'UTR-A$_{34}$*. 10 μM RO-3306 was added to the medium to prolong prophase arrest. **l** Representative stills from time-lapse movies of DDX6/LSM14B-depleted and DDX6/LSM14B/PABPC1L/PABPC4-depleted mouse oocytes expressing *mClove3-Ccnb1 3'UTR-A$_{34}$*. **m** Quantification and normalization of the mean fluorescence intensity of mClover3 in (**l**). Data are from two (**b, e, g–m**) or three (**d, f**) independent experiments. The number of analyzed oocytes is specified in italics. Data are shown as mean ± SD. *P*-values were calculated using unpaired two-tailed Student's *t* test (**f, g, h**) or two-way ANOVA with Tukey's *post-hoc* test (**m**). *P*-values are designated as ****$P < 0.0001$. Nonsignificant values are indicated as N.S. Scale bars, 10 μm. Source data are provided as a Source Data file.

length between the other two groups, was not significantly altered (Fig. 4f, h). Moreover, acute depletion of PABPCs did not significantly change mRNA stability or poly(A) tail length (Supplementary Fig. 4d-l). These data complement well with the reporter assay and also support that acute depletion of PABPCs causes a shift of translation from long-tailed mRNAs to tailless and short-tailed mRNAs.

Taken together, PABPCs may regulate translation in two ways: 1) they directly promote the translation of an mRNA through binding to its poly(A) tail as previously reported; 2) they indirectly repress the translation of tailless or short-tailed mRNAs by sequestering the translation machinery on long-tailed mRNAs. This mechanism ensures both active translation of long-tailed mRNAs and dormancy of tailless or short-tailed mRNAs. This also explains why acute depletion of PABPCs did not decrease, but increased translation of the cyclin B1 reporter mRNA with a short poly(A) tail (Fig. 3l, m).

## PABPCs indirectly repress cyclin B1 translation to maintain prophase arrest

Endogenous cyclin B1 mRNA also contains a short poly(A) tail in prophase (Fig. 3a, b). We therefore tested whether acute depletion of PABPCs increased its translation by performing the modified RIBOmap assay, and also analyzed how it affected prophase arrest. Acute depletion of PABPC1L slightly increased cyclin B1 translation (Fig. 5b), but this was not sufficient to cause premature meiotic resumption (Fig. 5c). Acute depletion of PABPC4 had no obvious effect on cyclin B1 translation and prophase arrest (Fig. 5b, c). By contrast, co-depletion of PABPC1L and PABPC4 significantly increased cyclin B1 translation without affecting its stability or poly(A) tail length and therefore caused premature meiotic resumption (Fig. 5a–c and Supplementary Fig. 5a–c). This result was further confirmed by the combination of *pabpc4* RNAi and PABPC1L Trim-Away (Supplementary Fig. 5d–f). Thus, PABPCs are also essential for maintaining cyclin B1 dormancy and prophase arrest. In contrast to DDX6 and LSM14B, which directly repress cyclin B1 translation, PABPCs may indirectly repress the translation of poly(A)-tail-less or short-tailed mRNAs, including cyclin B1, by sequestering the translation machinery on long-tailed mRNAs (Fig. 4). Although acute depletion of PABPCs showed weaker phenotypes than acute depletion of DDX6 and LSM14B (Fig. 1f, Fig. 2f, g and Fig. 5a–c), PABPC Trim-Away further enhanced the phenotypes of DDX6/LSM14B Trim-Away in terms of cyclin B1 translation (Fig. 3k–m) and the speed of premature meiotic resumption (Supplementary Fig. 5g-i).

Cyclin B1 mRNA is translationally repressed during prophase arrest and is translationally activated through poly(A) tail elongation when oocytes resume meiosis[35,36]. To investigate how PABPCs regulate cyclin B1 translation at this important transition, we used the reporter mRNA containing the cyclin B1 3'UTR but no poly(A) tail both in prophase and during oocyte meiotic maturation (Fig. 5d). When oocytes were arrested in prophase by supplementation with RO-3306, acute

depletion of PABPCs resulted in increased translation of the reporter mRNA (Fig. 5e–g), consistent with the data above (Fig. 4a–d and Supplementary Fig. 4a, b). When RO-3306 was withdrawn to allow oocytes to mature, the rate of translation was higher in the first 6 hours but lower in the next 6 hours in oocytes depleted of PABPCs compared to control oocytes (Fig. 5e, h, i, j). It is important to note that cyclin B1 poly(A) tail elongation occurs gradually during meiotic maturation[39]. Thus, in the first few hours, when the poly(A) tail was short, depletion of PABPCs promoted translation, whereas in the next few hours, when the poly(A) tail was long, depletion of PABPCs impaired translation. In all conditions, reporter mRNA levels were similar (Supplementary Fig. 5j). Together, these data provide further evidence that PABPCs regulate translation in two ways: 1) directly promoting the translation of an mRNA through binding to its poly(A) tail, 2) indirectly repressing the translation of tailless or short-tailed mRNAs by sequestering the translation machinery on long-tailed mRNAs.

## PABPCs and DDX6/LSM14B counteract each other on the same mRNA

Previous studies have focused on either translational activators or repressors, but it is unclear how they coordinate to regulate translation when they coexist on an mRNA, as in the case of cyclin B1, which is regulated by both PABPCs and DDX6/LSM14B (Figs. 2, 3, 5).

To systematically investigate the relationship between PABPCs and DDX6/LSM14B in the translational control of maternal mRNAs, we designed a reporter mRNA assay that allowed us to recruit defined numbers of PABPCs and DDX6/LSM14B proteins and to assess their effect on translation. To this end, we utilized the PP7 hairpin-PP7 coat protein (PCP) interaction[44]. By varying the number of PP7s and the length of poly(A) tails (Supplementary Fig. 6a, b), different numbers of DDX6/LSM14B and PABPCs were recruited to the reporter mRNA (Fig. 6a,c,e). The injection control *mScarlet* and the reporter mRNA *mClover3* containing several copies of PP7 and different lengths of poly(A) tails were co-injected with the tandem dimer of PCP (*tdPCP*), *tdPCP-Ddx6*, or *tdPCP-Lsm14b* (Fig. 6a, c, e). Translation of the reporter mRNA was assessed by the signal ratio of mClover3 to mScarlet.

When the reporter mRNA contained a short poly(A) tail, two DDX6 or LSM14B proteins were sufficient to repress its translation to a low level (Fig. 6a, b and Supplementary Fig. 6c), mimicking DDX6/LSM14B-mediated translational repression of endogenous cyclin B1 in prophase-arrested oocytes. However, when the reporter mRNA contained a long poly(A) tail, its translation was not efficiently repressed by two DDX6 or LSM14B proteins (Fig. 6c, d and Supplementary Fig. 6d). Notably, repression of the long-tailed reporter mRNA was enhanced by increasing the number of PP7s to recruit more DDX6 or LSM14B proteins (Fig. 6e, f and Supplementary Fig. 6e). Thus, PABPCs and DDX6/LSM14B counteracted each other in a quantity-dependent manner to determine translation efficiency.

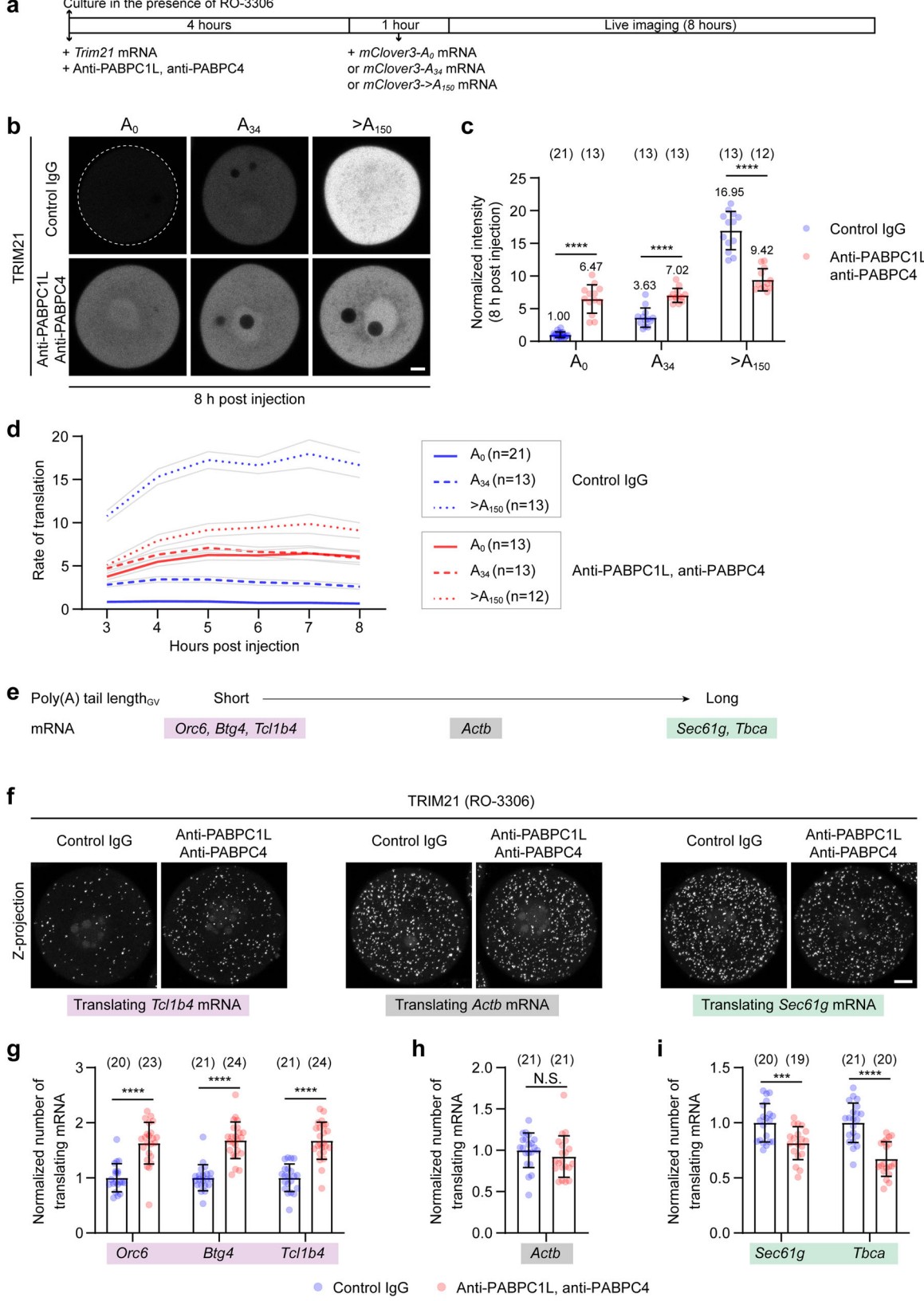

**Acute depletion of DDX6/LSM14B or PABPCs has distinct effects on oocyte meiotic maturation**

Finally, we examined how acute depletion of DDX6/LSM14 or PABPCs affected oocyte meiotic maturation. Acute depletion of DDX6 and LSM14B did not affect the extrusion of the first polar body (Fig. 1e, g). We then performed live-cell imaging to analyze oocyte meiotic

maturation in detail. Compared to control oocytes that underwent maturation after withdrawal of dbcAMP, oocytes depleted of DDX6 and LSM14B, which resumed meiosis in the presence of dbcAMP, did not show noticeable defects in spindle assembly and chromosome alignment in both meiosis I and II (Fig. 7a, Supplementary Fig. 7a, b and Supplementary Movie 1). Moreover, acute depletion of DDX6 and

**Fig. 4 | Acute depletion of PABPCs has opposite effects on translation of mRNAs with long or short poly(A) tails. a** Schematic diagram of the experiment in (**b**). Acute depletion of PABPC1L and PABPC4 was performed, followed by injection of *mClover3* reporter mRNAs with different poly(A) tail lengths. 10 μM RO-3306 was added to the medium to prolong prophase arrest. **b** Representative fluorescence images of control and PABPC1L/PABPC4-depleted mouse oocytes expressing *mClover3-A$_0$*, *mClover3-A$_{34}$*, or *mClover3->A$_{150}$* at 8 h after injection of these reporter mRNAs. **c** Quantification and normalization of the mean fluorescence intensity of mClover3 in (**b**). **d** The rate of translation over time was quantified by measuring the increase in fluorescence intensity every hour. **e** Endogenous mRNAs with different poly(A) tail lengths. **f** Representative RIBOmap images of translating mRNAs in control and PABPC1L/PABPC4-depleted oocytes. Oocytes were incubated for 6 h in the presence of 10 μM RO-3306 after injection of *Trim21* mRNA and antibodies. **g**–**i** Quantification and normalization of the number of translating mRNAs in (**f**). Data are all from two independent experiments. The number of analyzed oocytes is specified in italics. Data are shown as mean ± SD (**c**, **g**, **h**, **i**) or mean ± SEM (**d**). *P*-values were calculated using two-way ANOVA with Tukey's *post-hoc* test (**c**) or unpaired two-tailed Student's *t* test (**g**, **h**, **i**). *P* values are designated as ****$P < 0.0001$, ***$P < 0.001$. Nonsignificant values are indicated as N.S. Scale bars, 10 μm. Source data are provided as a Source Data file.

LSM14B resulted in more rapid progression of meiosis I (Supplementary Fig. 7c). Thus, although DDX6 and LSM14B are important for normal oocyte growth and prophase arrest, they are not directly involved in spindle assembly, chromosome alignment and segregation during oocyte meiotic maturation.

By contrast, acute depletion of PABPC1L and PABPC4 caused obvious defects in oocyte meiotic maturation (Fig. 7b–g and Supplementary Movie 2). In oocytes depleted of PABPCs, the progression of meiosis I was slower (Fig. 7c, d), the morphology of the MI spindle was abnormal (Fig. 7c, e), and the MI spindle contained misaligned chromosomes (Fig. 7c, f). Moreover, oocytes showed severe defects in the assembly of the MII spindle (Fig. 7c, g). Thus, PABPCs are essential for oocyte meiotic maturation. Importantly, when DDX6 and LSM14B were depleted just before meiotic resumption (Fig. 7b), the meiosis I defects caused by PABPC depletion were largely rescued (Fig. 7c–f and Supplementary Movie 2), whereas the defects in meiosis II remained (Fig. 7c, g and Supplementary Movie 2). These data demonstrate the counteraction between PABPCs and DDX6/LSM14B at the physiological level.

## Discussion

Our study demonstrates that prophase arrest in mouse oocytes is regulated at the translational level and establishes a mechanistic framework for this regulation, centered on translational repression of dormant cyclin B1 mRNAs. Cyclin B1 is one of the most important maternal mRNAs that accumulate during oocyte growth. We identified two distinct mechanisms for repressing the translation of stored cyclin B1 mRNAs, which is essential for maintaining prophase arrest. On the one hand, DDX6 and LSM14B directly repress cyclin B1 translation, probably by forming a complex with CPEB1 that binds to the 3'UTR of cyclin B1 (Fig. 8a). On the other hand, PABPCs indirectly repress the translation of poly(A)-tail-less or short-tailed mRNAs, including cyclin B1, by sequestering the translation machinery on long-tailed mRNAs (Fig. 8a). These two mechanisms ensure that the oocyte safely accumulates large amounts of cyclin B1 mRNA during growth without causing premature resumption of meiosis, and thus allows the oocyte to grow to its full size.

We found that acute depletion of PABPCs results in a shift of translation events from long-tailed mRNAs to tailless and short-tailed mRNAs, so that the translation of all mRNAs becomes similar, irrespective of their poly(A) tail length (Fig. 8a). This supports a mechanism whereby the silencing of tailless and short-tailed mRNAs is partly due to the sequestration of the translation machinery on long-tailed mRNAs. PABPCs interact with EIF4G to promote mRNA circularization, which enhances translation by recycling ribosomes[40,45]. It is possible that in the presence of PABPCs, ribosomes are predominantly occupied by long-tailed circularized mRNAs, whereas in the absence of PABPCs, linearized mRNAs all have an equal chance of acquiring ribosomes (Fig. 8a).

It is worth noting that the poly(A) tail is not the only factor that determines translation efficiency. Mounting evidence suggests that cis-elements in the 3'UTR and their corresponding trans-factors also play an important role in translational control[35,41,46–49]. Our findings indicate that this is also the case for cyclin B1 mRNA, translation of

which is regulated by both DDX6/LSM14B and PABPCs. An intriguing question is how these factors coordinate to regulate the translation of an mRNA. Through a series of tethering experiments, we discovered that PABPCs bound to the poly(A) tail and DDX6/LSM14B bound to the 3'UTR counteract each other in a quantity-dependent manner to determine translation efficiency (Fig. 6). 4E-T, which interacts with DDX6/LSM14B, and EIF4G, which interacts with PABPCs, have been proposed to compete for binding to EIF4E[50]. Therefore, the DDX6/LSM14B/4E-T complex and the PABPC/EIF4G complex may compete in a quantity-dependent manner to regulate the translation of the bound mRNA (Fig. 8b).

DDX6 and LSM14B effectively repress cyclin B1 translation in prophase-arrested oocytes. How is the repressive effect alleviated to allow cyclin B1 translation when oocytes resume meiosis? A longer poly(A) tail diminishes DDX6/LSM14B-mediated translational repression (Fig. 6c, d), suggesting that poly(A) tail elongation itself serves as a potential mechanism for overcoming DDX6/LSM14B-mediated translational repression when oocytes resume meiosis. Other mechanisms may also be involved. CPEB1 is phosphorylated and degraded during oocyte meiotic maturation[36,51,52]. DDX6, LSM14B and 4E-T are also phosphorylated[9], although their protein levels remain relatively unchanged (Fig. 2a, b). Whether these events are linked to the release of DDX6/LSM14B-mediated translational repression remains unclear.

We observed that acute depletion of DDX6 and LSM14B by Trim-Away, in contrast to gene knockout of *Ddx6* or *Lsm14b*[20–24], results in a range of distinct phenotypes in late-stage mouse oocytes. For instance, acute removal of DDX6 and LSM14B does not affect MII spindle formation, whereas *Lsm14b* gene knockout results in pronucleus formation during meiosis II. This discrepancy could be attributed to the fact that knockout of *Ddx6* or *Lsm14b* also impacts early oogenesis[20,23], which ultimately leads to defects in meiosis II. Future studies could combine gene knockout and Trim-Away approaches to gain a deeper understanding of the functions of RNA-binding proteins at different stages of oocyte and embryo development. While DDX6/LSM14B and PABPCs regulate prophase arrest mainly through cyclin B1, the mechanisms by which they regulate translation also apply to other mRNAs. Future research should combine Trim-Away with the recently developed Ribo-seq techniques for low-input samples to reveal the full translatome they regulate[41].

## Methods

### Preparation and culture of mouse oocytes and follicles

Maintenance and handling of all mice were carried out in the animal facility of Max Planck Institute for Multidisciplinary Sciences according to international animal welfare rules (Federation for Laboratory Animal Science Associations guidelines and recommendations). Requirements of formal control of the German national authorities and funding organizations were satisfied, and the study received approval by the Niedersächsisches Landesamt für Verbraucherschutz und Lebensmittelsicherheit (LAVES). C57BL/6 N and CD1 mice were maintained in a specific pathogen-free environment according to The Federation of European Laboratory Animal Science Associations guidelines and recommendations. Full-grown oocytes with a centered germinal vesicle were isolated from ovaries of 7- to 10-week-old mice

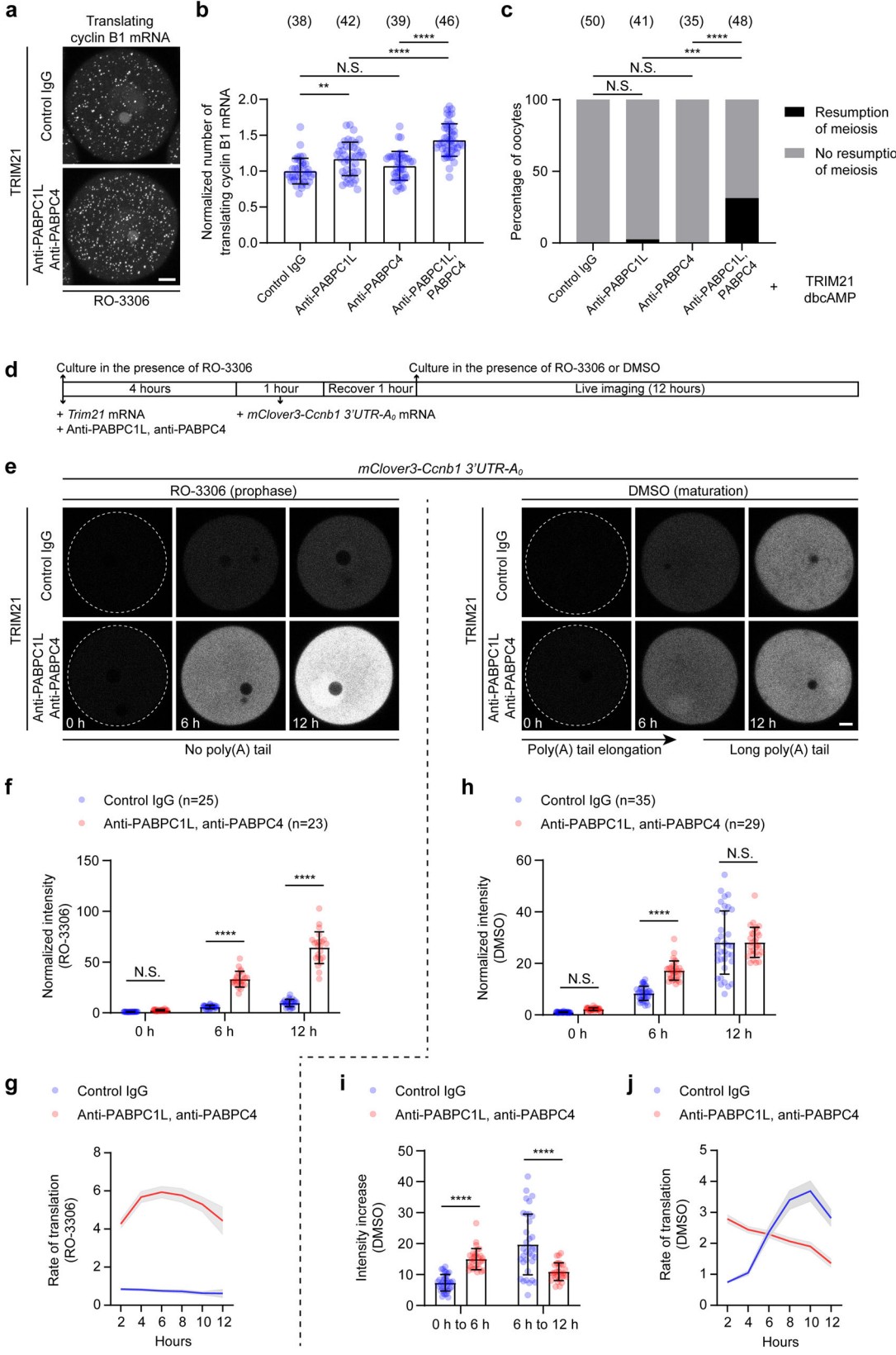

and were kept arrested in prophase in homemade phenol red-free M2 medium supplemented with 250 μM dbcAMP under paraffin oil (NidaCon #NO-400K).

Follicles were mechanically isolated from 14-day-old CD1 female mice in HEPES-buffered Minimal Essential Medium (MEM) with Gluta-MAX (Thermo Fisher Scientific #42360024) supplemented with 5%

Fetal Bovine Serum (FBS) (Thermo Fisher Scientific #16000044) and 0.1× Penicillin G/Streptomycin. Compact follicles with a centered oocyte and several layers of granulose cells were collected for micro-injection and then cultured in MEM α with nucleosides and GlutaMAX (Thermo Fisher Scientific #32571028) supplemented with 5% FBS, 0.01 μg/ml ovine Follicle Stimulating Hormone (National Hormone

**Fig. 5 | Acute depletion of PABPCs causes increased cyclin B1 translation and premature resumption of meiosis. a** Representative RIBOmap images of translating cyclin B1 mRNAs in control and PABPC1L/PABPC4-depleted oocytes. Oocytes were incubated for 6 h in the presence of 10 μM RO-3306 after injection of *Trim21* mRNA and antibodies. **b** Quantification and normalization of the number of translating cyclin B1 mRNAs in conotrol, PABPC1L-depleted, PABPC4-depleted and PABPC1L/PABPC4-depleted oocytes. Oocytes were incubated for 6 h in the presence of 10 μM RO-3306 after injection of *Trim21* mRNA and antibodies. **c** Percentage of mouse oocytes that resume meiosis in 18 h in the presence of dbcAMP. PABPC1L and PABPC4 were acutely depleted individually or together. **d** Schematic diagram of the experiment in (**e**). Acute depletion of PABPC1L and PABPC4 was performed, followed by injection of the *mClove3-Ccnb1 3'UTR-$A_O$* reporter mRNA. Oocytes were then imaged in the presence (for prolonging prophase arrest) or absence (for maturation) of 10 μM RO-3306. **e** Representative stills from time-lapse movies of control and PABPC1L/PABPC4-depleted mouse oocytes expressing *mClove3-Ccnb1 3'UTR-$A_O$*. Oocytes were incubated in the presence of 10 μM RO-3306 to prolong prophase arrest or in the absence of RO-3306 to promote poly(A) tail elongation of the reporter mRNA. **f** Quantification and

normalization of the mean fluorescence intensity of mClover3 in prophase-arrested oocytes. **g** The rate of translation of the reporter mRNA in prophase-arrested oocytes was quantified by measuring the increase in fluorescence intensity every hour. **h** Quantification and normalization of the mean fluorescence intensity of mClover3 in maturing oocytes. **i** Comparison of the translation of the reporter mRNA in the first 6 h and in the second 6 h during oocyte meiotic maturation and poly(A) tail elongation. Data were calculated by subtracting the 0 h intensity from the 6 h intensity (translation in the first 6 h) or by subtracting the 6 h intensity from the 12 h intensity (translation in the second 6 h). **j** The rate of translation of the reporter mRNA in maturing oocytes was quantified by measuring the increase in fluorescence intensity every hour. Data are from three (**a**, **b**) or two (**c**–**j**) independent experiments. The number of analyzed oocytes is specified in italics. Data are shown as mean ± SD (**b**, **f**, **h**, **i**) or mean ± SEM (**g**, **j**). *P*-values were calculated using one-way (**b**) or two-way (**f**, **h**, **i**) ANOVA with Tukey's *post-hoc* test or two-tailed Fisher's exact test (**c**). *P* values are designated as \*\*\*\**P* < 0.0001, \*\*\**P* < 0.001, \*\**P* < 0.01. Nonsignificant values are indicated as N.S. Scale bars, 10 μm. Source data are provided as a Source Data file.

and Peptide Program #NIDKK-oFSH-20), 1× Insulin/Transferrin/Sodium selenite (Thermo Fisher Scientific #41400045), and 0.1× Penicillin G/Streptomycin on collagen-coated inserts (Corning #3491/3493) at 37 °C and 5% CO2 as previously described[53]. Medium surrounding the insert was exchanged every 3 days. After 9 days of culture, oocytes were harvested in modified M2 medium with 10% FBS instead of 4 mg/ml bovine serum albumin (BSA).

## Immunofluorescence

Oocytes were fixed with a fixation solution containing 100 mM Hepes (pH 7.0, titrated with KOH), 50 mM EGTA (pH 7.0, titrated with KOH), 10 mM $MgSO_4$, and 2% methanol-free formaldehyde for 30 min at 37 °C. Fixed oocytes were washed and permeabilized with PBT buffer (0.5% Triton X-100 in PBS) for 1 h at room temperature or overnight at 4 °C. Permeabilized oocytes were blocked with PBT-BSA buffer (PBS containing 3% BSA and 0.1% Triton X-100) for 1 h at room temperature or overnight at 4 °C. Oocytes were then sequentially incubated with primary and secondary antibodies in PBT-BSA for 1.5 h each at room temperature. Extensive washing with PBT-BSA was performed in both steps. Hoechst 33342 (Thermo Fisher Scientific #H3570) was used for DNA labeling together with secondary antibodies. Primary antibodies used were mouse anti-DDX6 (Sigma-Aldrich #SAB4200837), rabbit anti-LSM14B (Thermo Fisher Scientific #PA5-66371), rabbit anti-4E-T (Thermo Fisher Scientific #PA5-51680), rabbit anti-YBX2 (Abcam #ab33164), and guinea pig anti-ZAR1 (made by Cambridge Research Biochemicals)[9]. Secondary antibodies used were Alexa Fluor 488-, 568- or 647-conjugated goat anti-mouse, rabbit or guinea pig IgG (highly) cross-adsorbed secondary antibodies (Thermo Fisher Scientific). All antibodies were diluted at 1:100.

## Modified RIBOmap

The assay was performed as previously described[37] with some modifications for application to oocytes. Fluorescent probes instead of in situ sequencing were used directly to label the amplicons.

Briefly, oocytes were fixed with 2% methanol-free formaldehyde in PBS for 30 min at 37 °C. After extensive washing with PBS containing 0.05 mg/ml BSA (Thermo Fisher Scientific #AM2616) and 0.04 U/μl SUPERase•In RNase Inhibitor (Thermo Fisher Scientific #AM2696), oocytes were permeabilized with fresh 70% ethanol for 20 min at room temperature. Oocytes were then washed three times with Hybridization Buffer consisting of 2× SSC, 10% formamide, 0.1 mg/ml yeast tRNA (Thermo Fisher Scientific #AM7119), 0.2 U/μl SUPERase•In RNase Inhibitor, and 0.1% Triton X-100. Hybridization Solution was prepared by diluting all probes, including 5 Splints, 4 mRNA-specific Padlocks and 4 mRNA-specific Primers, to a final concentration of 50 nM with Hybridization Buffer. Oocytes were incubated with Hybridization

Solution for 4-6 h at 40 °C in a 6-well dish (Agtech, #3926909910) with some wet tissue paper around. After hybridization, oocytes were washed twice with PBSTR (0.1% Triton X-100 and 0.1 U/μl SUPERase•In RNase Inhibitor in PBS) and once with 4× SSC diluted with PBSTR. All washes were performed at 37 °C for 10 min. Oocytes were then rinsed briefly with PBSTR and incubated for 1.5 h at 25 °C with Ligation Solution consisting of 0.25 Weiss U/μl T4 DNA ligase (Thermo Fisher Scientific #EL0012) in 1× buffer, 0.5 mg/ml BSA, and 0.4 U/μl of SUPERase•In RNase Inhibitor as previously described[37]. After two washes with PBSTR for 5 min each, oocytes were incubated with Amplification-Labeling Mixture consisting of 0.5 U/μl Phi29 DNA polymerase (NEB #M0269S) in 1× buffer, 250 μM dNTP (Thermo Fisher Scientific #18427013), 0.5 mg/ml BSA, 0.4 U/μl of SUPERase•In RNase Inhibitor, 1 μM Detection Probe 1, and 1 μM Detection Probe 2 for 2 h at 30 °C. Oocytes were subsequently washed twice with PBSTR-BSA (0.1% Triton X-100, 0.04 U/μl SUPERase•In RNase Inhibitor and 0.05 mg/ml BSA in PBS) for 5 min each at room temperature and stored in the same buffer at 4 °C for no longer than 24 h before imaging. The Splints and some mRNA-specific Padlocks/Primers were designed by the previous study[37]. Other mRNA-specific Padlocks/Primers were designed using Picky 2.2 software[54] as previously described[37]. Only the hybridization regions of Padlocks/Primers need to be replaced when designing new Padlocks/Primers for new mRNAs. The two fluorophore-labeled Detection Probes complement with all the amplicons. The sequences of all probes are listed in Supplementary Dataset 1.

## Expression constructs and mRNA synthesis

Primers used for plasmid construction are shown in Supplementary Dataset 2. To construct pGEMHE-CGH-1-mClover3, *C. elegans cgh-1* was synthesized (Integrated DNA Technologies), and assembled with linearized pGEMHE-mClover3-N1 (NheI/NcoI) through Gibson Assembly (NEB #E2621S). To build pGEMHE-mClover3-*Ccnb1* 3'UTR with a short poly(A) tail ($A_{34}$), the longest *Ccnb1* 3'UTR was amplified from a mouse oocyte cDNA library using primers P1/P2, and was assembled with linearized pGEMHE-mClover3-N1 (XbaI) containing the short poly(A) tail through Gibson Assembly. To build pGEMHE-mClover3-*Ccnb1* 3'UTR without poly(A) tail ($A_0$), the longest *Ccnb1* 3'UTR was amplified from a mouse oocyte cDNA library using primers P1/P3, and was assembled with linearized pGEMHE-mClover3-C1 (XhoI/AscI) without poly(A) tail by Gibson assembly. To construct pGEMHE-mClover3-10×PP7, PP7 repeat was amplified from phage-CMV-CFP-24×PP7 (Addgene #40652)[55] using primers P4/P5 and assembled with pGEMHE-mClover3-N1 (XbaI) through Gibson Assembly. Sequencing was performed to get the clone containing 10×PP7. Other constructs used in this study include pGEMHE-TRIM21[56], pGEMHE-mClover3-C1[9], pGEMHE-mScarlet-C1[9], pGEMHE-DDX6-mScarlet[9], pGEMHE-LSM14B-

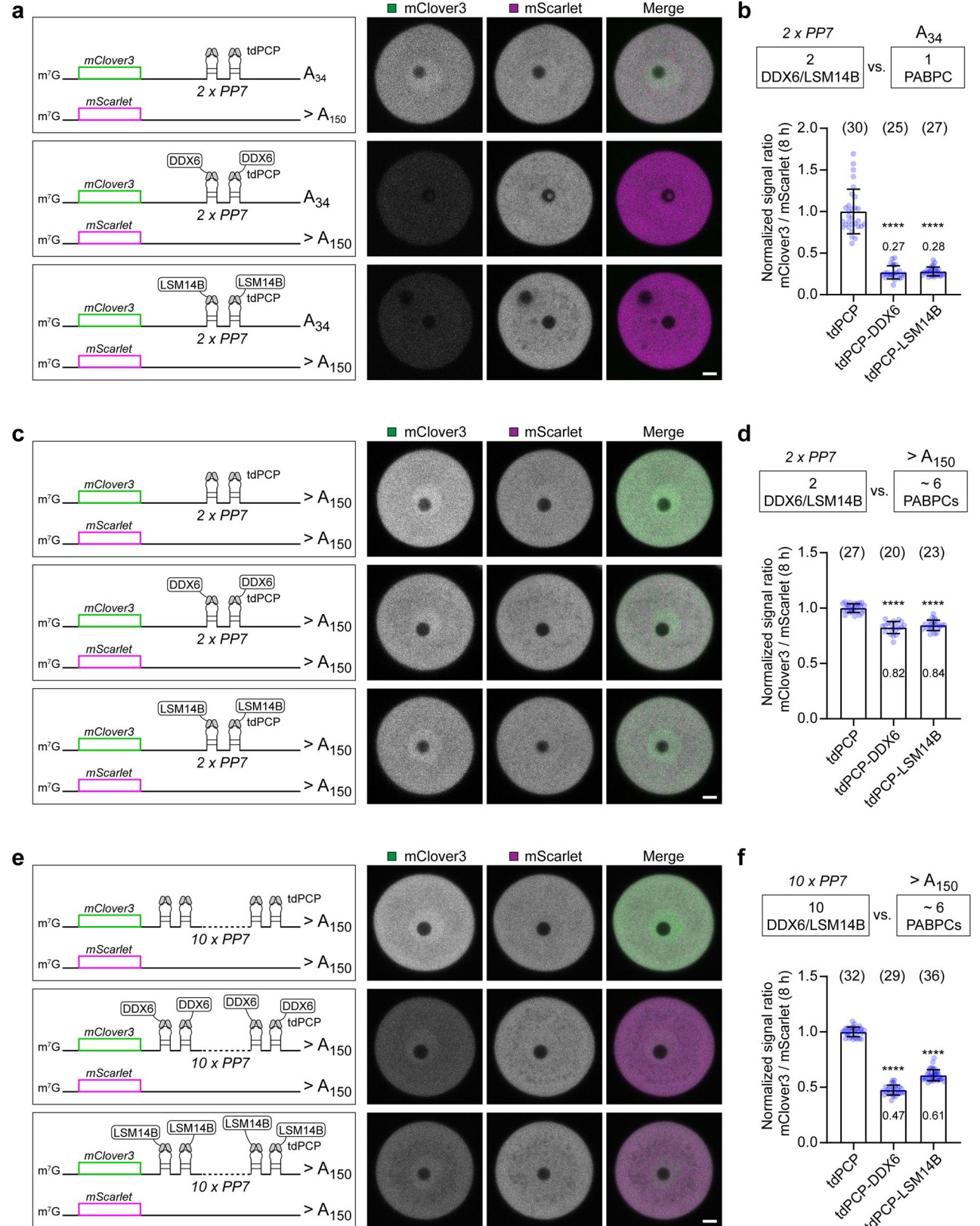

**Fig. 6 | PABPCs and DDX6/LSM14B counteract each other on the same mRNA.**
**a**, **c**, **e** Representative fluorescence images of mouse GV oocytes expressing *mClover3* with different numbers of PP7s and different lengths of poly(A) tails, *mScarlet*, and *tdPCP*, *tdPCP-Ddx6* or *tdPCP-Lsm14b* at 8 h after mRNA injection. Schematic diagrams of the experiments are shown on the left. Green, mClover3; magenta, mScarlet. **b**, **d**, **f** Quantification and normalization of the fluorescence intensity ratio of mClover3 to mScarlet in (**a**, **c**, **e**). Data are all from two independent experiments. The number of analyzed oocytes is specified in italics. Data are shown as mean ± SD. *P*-values were calculated using one-way ANOVA with Tukey's *post-hoc* test. *P* values are designated as ****$P < 0.0001$. Scale bars, 10 μm. Source data are provided as a Source Data file.

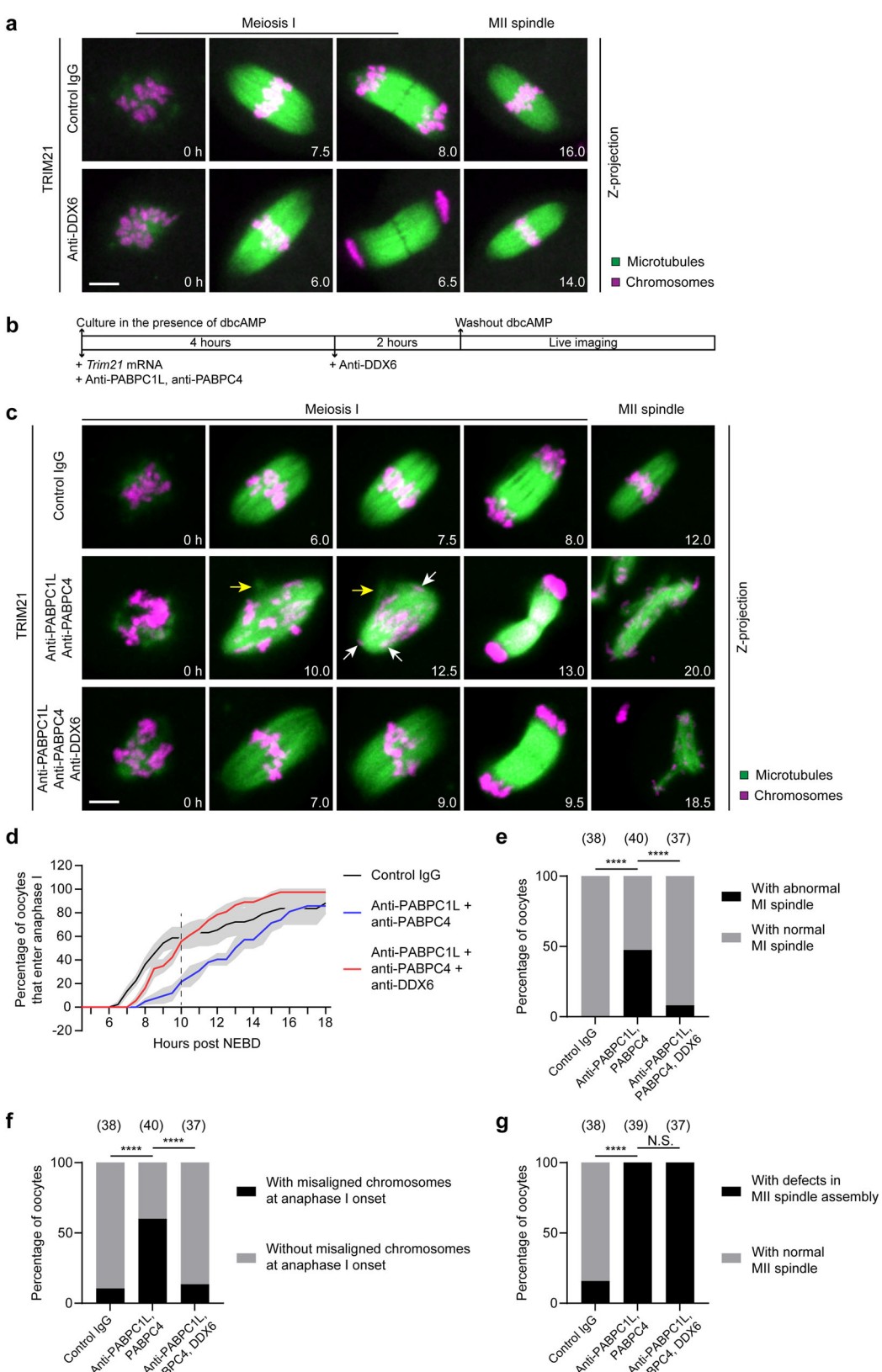

mScarlet[9], pGEMHE-Cyclin B1-mCherry[57], pGEMHE-mClover3-2×PP7[9], pGEMHE-mClover3-MAP4-MTBD[56], and pGEMHE-H2B-mRFP[58]. All mRNAs were synthesized using the HiScribe T7 ARCA mRNA Kit (with tailing) (NEB #E2060S), and then were purified with the RNeasy Mini Kit (Qiagen #74104). All mRNAs contain *Xenopus* globin 5′UTR and 3′UTR unless specified.

### RNA interference

All short interfering RNAs (siRNAs) were purchased from Thermo Fisher Scientific (#4390771). Sequences of siRNAs are shown in Supplementary Dataset 3. For *Ccnb1* knockdown, equal amounts of 100 µM *Ccnb1* siRNA 1 (Thermo Fisher Scientific #s114071) and *Ccnb1* siRNA 2 (Thermo Fisher Scientific #s114072) were mixed, and 4 pl of

**Fig. 7 | Acute depletion of DDX6/LSM14B or PABPCs has distinct effects on oocyte meiotic maturation. a** Representative stills from time-lapse movies of control and DDX6/LSM14B-depleted mouse oocytes. Green, microtubules (mClover3-MAP4-MTBD); magenta, chromosomes (H2B-mRFP). Time is indicated as hours after NEBD. **b** Schematic diagram of the experiment in (**c**). Acute depletion of PABPC1L and PABPC4 was performed, followed by acute depletion of DDX6 and LSM14B. **c** Representative stills from time-lapse movies of control, PABPC1L/PABPC4-depleted, and PABPC1L/PABPC4/DDX6/LSM14B-depleted mouse oocytes. Green, microtubules (mClover3-MAP4-MTBD); magenta, chromosomes (H2B-mRFP). Time is indicated as hours after NEBD. Yellow arrows highlight abnormal

protrusions of spindles. White arrows point to misaligned chromosomes at anaphase I onset. **d** Quantification of the time from NEBD to chromosome separation in (**c**). **e** Percentage of oocytes with abnormal MI spindle in (**c**). **f** Percentage of oocytes with misaligned chromosomes at anaphase I onset in (**c**). **g** Percentage of oocytes with defects in MII spindle assembly in (**c**). Data are from two (**a**) or three (**b**–**g**) independent experiments. The number of analyzed oocytes is specified in italics. Data are shown as mean ± SEM (**d**). P values were calculated using two-tailed Fisher's exact test (**e**, **f**, **g**). P values are designated as ****$P < 0.0001$. Nonsignificant values are indicated as N.S. Scale bars, 10 μm. Source data are provided as a Source Data file.

the mixture or the same amount of control siRNA (Thermo Fisher Scientific #4390843) was injected into full-sized oocytes. Oocytes were incubated for 24 h before further experiments. For knockdown of *Pbapc1* and *Pabpc4*, equal amounts of 100 μM *Pbapc1* siRNA 1 (Thermo Fisher Scientific #s71211), *Pbapc1* siRNA 2 (Thermo Fisher Scientific #s71213), *Pbapc4* siRNA 1 (Thermo Fisher Scientific #s106566), and *Pbapc4* siRNA 2 (Thermo Fisher Scientific #s106567) were mixed, and 4 pl of the mixture or the same amount of control siRNA was injected into follicles collected from 14-day-old female mice. The follicles were cultured for 9 days before further experiments. PABPC4, but not PABPC1, was successfully knocked down in this experiment.

### Trim-Away in mouse oocytes

Ultrafiltration (Merck Millipore #UFC505024) was performed to change the buffer of all antibodies and control IgG (Merck Millipore #12-370, #12-371) to PBS. 2 pl of mouse *Trim21* mRNA and 2 pl of antibody containing 0.1% NP-40 were co-injected as previously described[59]. Injected oocytes were kept in M2 medium supplemented with 250 μM dbcAMP or 10 μM RO-3306. Mouse *Trim21* mRNA was injected at a concentration of 0.5 μM or 1 μM in the injection solution. Mouse anti-DDX6 (Sigma-Aldrich #SAB4200837), rabbit anti-DDX6 (Thermo Fisher Scientific #A300-461A), rabbit anti-CPEB1 (Proteintech #13274-1-AP), mouse anti-PABPC1L (Santa Cruz Biotechnology #sc-515476L), and rabbit anti-PABPC4 (Thermo Fisher Scientific #PA5-96483) were all injected at a concentration of 2 mg/ml in the injection solution.

### Translation of mRNAs with different poly(A) lengths

The *mClover3* reporter mRNA containing a short poly(A) tail ($A_{34}$) was directly transcribed using the linearized pGEMHE-mClover3-2×PP7 construct as the template. Poly(A) tailing of the mRNA was performed to get *mClover3* reporter mRNA with a poly(A) tail longer than 150 bp (>$A_{150}$). To obtain *mClover3* reporter mRNA without poly(A) tail ($A_0$), the DNA fragment containing the T7 promoter, *mClover3* coding sequence, and the entire 3′UTR was amplified from pGEMHE-mClover3-2×PP7. The poly(A) tail was removed during amplification. This DNA fragment was then used as the template to transcribe *mClover3* reporter mRNA without poly(A) tail. 4 pl of 1 μM *mClover3-$A_0$*, *mClover3-$A_{34}$* or *mClover3->$A_{150}$* mRNA was injected into oocytes and their expression was assessed at 3 h and 8 h after mRNA injection.

### tdPCP-PP7 tethering assay

The reporter mRNA *mClover3-2×PP7-$A_{34}$* was transcribed directly from the linearized pGEMHE-mClover3-2×PP7 construct or from a PCR fragment containing T7 promoter, *mClover3-2×PP7* and $A_{34}$. The reporter mRNA *mClover3-2×PP7->$A_{150}$* was obtained by poly(A) tailing of *mClover3-2×PP7-$A_{34}$*. The reporter mRNA *mClover3-10×PP7->$A_{150}$* was transcribed from the linearized pGEMHE-mClover3-10×PP7 construct followed by poly(A) tailing. 0.2 μM reporter mRNA, *mClover3-2×PP7-$A_{34}$*, *mClover3-2×PP7->$A_{150}$* or *mClover3-10×PP7->$A_{150}$*, and 0.1 μM injection control *mScarlet* were co-injected with either 0.5 μM *tdPCP* (tandem dimer of PCP), 0.5 μM *tdPCP-Ddx6* or 0.5 μM *tdPCP-Lsm14b* mRNA into prophase-arrested oocytes. The expression of

reporter mRNA was assessed by the signal ratio of mClover3 to mScarlet at 2 h and 8 h after mRNA injection.

### Labeling of spindle and chromosomes in live oocytes

To label spindle and chromosomes for live imaging, 2 pl of mRNA mixture containing 100 ng/μl *mClover3-Map4-MTBD* and 100 ng/μl *H2B-mRFP* mRNA was injected into prophase-arrested oocytes. Oocytes were incubated for 6 h before dbcAMP was withdrawn. *Trim21* mRNA and antibodies were co-injected to perform Trim-Away simultaneously.

### Vital stain labeling and drug treatment

5-SiR-Hoechst[60] was reconstituted in DMSO as a 1 mM stock. RO-3306 was reconstituted in DMSO as a 10 mM stock. To distinguish between SN and NSN oocytes, 125 nM 5-SiR-Hoechst was added to the culture medium throughout the experiment. In all experiments, oocytes were collected in M2 medium supplemented with 250 μM dbcAMP. In some Trim-Away experiments, oocytes were transferred to M2 medium containing 10 μM RO-3306 immediately after injection to prolong prophase arrest.

### Confocal microscopy

For confocal imaging of live oocytes, oocytes were imaged in 2 μl of M2 medium under paraffin oil in a 35 mm dish (MatTek # P35G-1.0-14-C) or in 800 μl of M2 medium without oil in a 4-chamber 35 mm dish (Greiner Bio-One #627871). For imaging of fixed oocytes, oocytes were imaged in 2 μl of 1% polyvinylpyrrolidone (IrvineScientific #90123) in PBS under paraffin oil in a 35 mm dish (MatTek # P35G-1.0-14-C). Images were acquired with LSM 880 confocal laser scanning microscopes (Zeiss) equipped with an environmental incubator box and a 40× C-Apochromat 1.2 NA water-immersion objective. Images of the control and experimental groups were acquired under identical imaging conditions on the same microscope. Movies were smoothened with a Gaussian filter (sigma = 1.3) in ZEN and aligned using Hyper-StackReg in Fiji (NIH)[61]. Care was taken that the imaging conditions (laser power, pixel-dwell time, and detector gain) did not cause phototoxicity (for live imaging), photobleaching, or saturation.

### Immunoblotting

Oocytes were rinsed with PBS, resuspended in 10 μl PBS, and immediately snap frozen in liquid nitrogen. 10 μl of 2×NuPAGE LDS Sample Buffer (Thermo Fisher Scientific #NP0007) containing 100 mM DTT was then added, followed by heating at 95 °C for 10 min. Samples were resolved on a 10-well NuPAGE 10% Bis-Tris protein gel of 1.0 mm thickness (Thermo Fisher Scientific #NP0301BOX) using NuPAGE MOPS SDS Running Buffer (Thermo Fisher Scientific #NP0001). Proteins were then transferred onto a 0.45 μm PVDF membrane (Thermo Fisher Scientific #LC2005) using NuPAGE Transfer Buffer (Thermo Fisher Scientific #NP0006). Blocking and antibody incubations were performed in PBS containing 5% skim milk and 0.1% Tween-20. Primary antibodies used were rabbit anti-DDX6 diluted at 1:2000 (Abcam #ab174277), rabbit anti-LSM14B diluted at 1:500 (Thermo Fisher Scientific #PA5-66371),

**a**

| Mechanism I: a complex containing DDX6, LSM14B and CPEB1 represses the translation of stored cyclin B1 mRNAs in prophase I-arrested oocytes | Mechanism II: PABPCs indirectly repress the translation of poly(A)-tail-less or short-tailed mRNAs, including cyclin B1, by sequestering the translation machinery on long-tailed mRNAs |

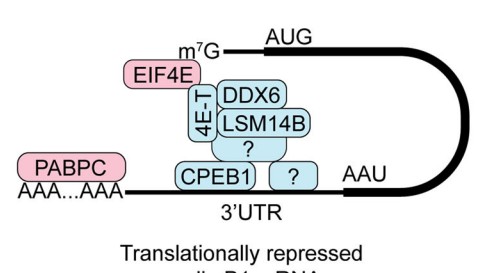

Translationally repressed
cyclin B1 mRNA

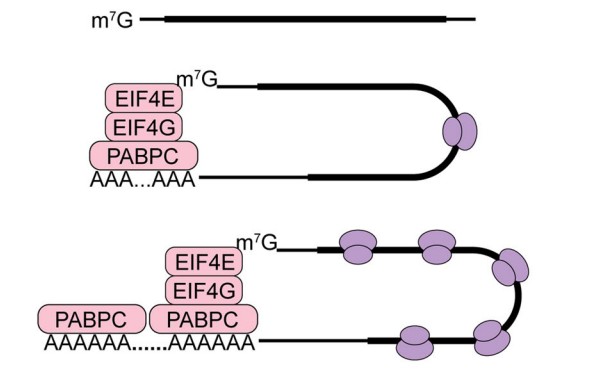

| Acute depletion of DDX6 and LSM14B leads to premature translation of stored cyclin B1 mRNAs, which impairs prophase I arrest | Acute depletion of PABPCs causes a shift of translation events from long-tailed mRNAs to tailless and short-tailed mRNAs |

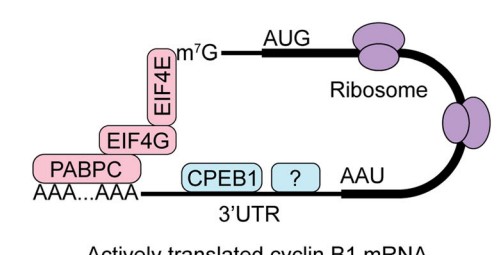

Actively translated cyclin B1 mRNA
(Independent of poly(A) tail elongation)

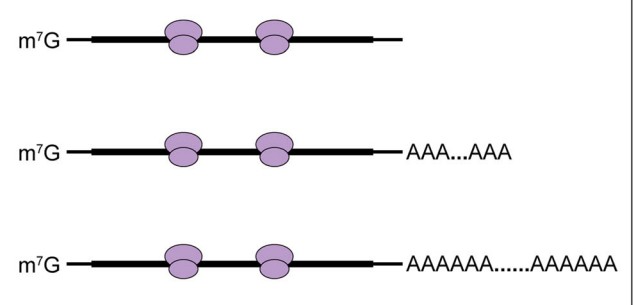

**b**

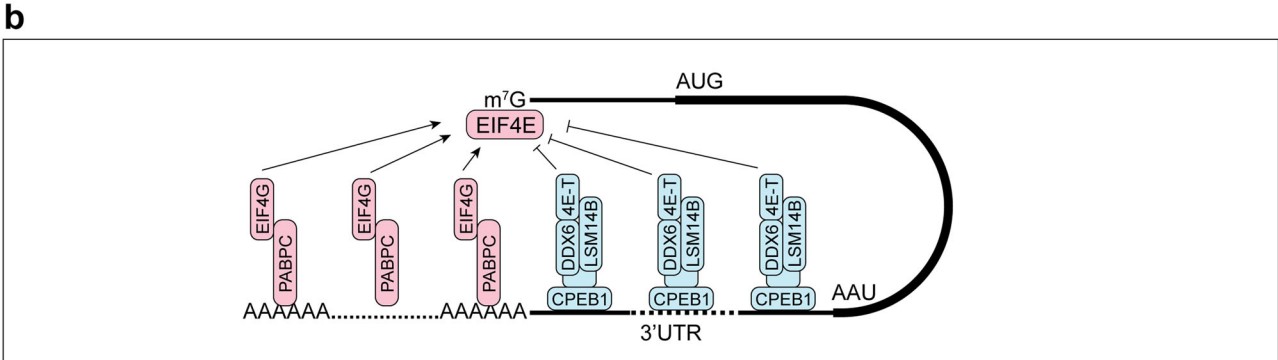

**Fig. 8 | Models showing two mechanisms that regulate mRNA dormancy and prophase I arrest. a** Two mechanisms acting through the 3′UTR and the poly(A) tail respectively maintain mRNA dormancy and prophase arrest. Mechanism I: a complex containing DDX6, LSM14B and CPEB1 directly represses the translation of stored cyclin B1 mRNAs by interacting with its 3′UTR to maintain prophase arrest. Mechanism II: PABPCs indirectly repress the translation of poly(A)-tail-less or short-tailed mRNAs, including cyclin B1, by sequestering the translation machinery on long-tailed mRNAs to maintain prophase arrest. **b** Translational activators (pink) and repressors (blue) on the same mRNA counteract each other in a quantity-dependent manner to determine translation efficiency.

rabbit anti-4E-T diluted at 1:500 (Thermo Fisher Scientific #PA5-51680), rabbit anti-DDB1 diluted at 1:5000 (Abcam #ab109027), mouse anti-cyclin B1 diluted at 1:200 (Santa Cruz Biotechnology #sc-245), rabbit anti-MOS diluted at 1:200 (Santa Cruz Biotechnology #sc-1093-R), mouse anti-CDK1 diluted at 1:200 (Thermo Fisher Scientific #MA5-11472), rabbit anti-phospho-CDK1(Tyr15) diluted at 1:1000 (Cell Signaling Technology #9111), rabbit anti-CPEB1 diluted at 1:1000 (Proteintech #13274-1-AP), mouse anti-PABPC1L diluted at 1:200 (Santa Cruz Biotechnology #sc-515476), rabbit anti-PABPC1 diluted at 1:1000 (Proteintech #10970-1-AP), and rabbit anti-PABPC4 diluted at 1:1000 (Thermo Fisher Scientific #PA5-96483). DDB-1 was used as an internal control. Secondary antibodies used were HRP-conjugated goat anti-mouse IgG (Abcam #ab205719) and goat anti-rabbit IgG (Abcam #ab205718). Blots were developed with SuperSignal West Femto Maximum Sensitivity Substrate (Thermo Fisher Scientific #34095) and documented with Amersham Imager

600 (Cytiva). Care was taken that the exposure time did not cause saturation. (See the Source Data).

## RT-qPCR

10 oocytes from the same mouse were pooled for total RNA purification using the RNeasy Micro Kit (Qiagen #74004). All RNA was reverse transcribed using the SuperScript IV First-Strand Synthesis System (Thermo Fisher Scientific #18091050). The cDNA was diluted 1:10, and 5 μl was used for a 25 μl qPCR reaction. qPCR was performed on the Rotor-Gene Q system (Qiagen) using the PowerUp SYBR Green Master Mix (Thermo Fisher Scientific #A25742). Endogenous actin (*Actb*), 18S rRNA or the co-injected *mScarlet* was used as an internal control to calculate relative transcript levels. Three biological replicates (mice) were included in each experiment. Primers used for qPCR are shown in Supplementary Dataset 4.

## Poly(A) tail length assay

Total RNA was extracted from 20 oocytes using the RNeasy Micro Kit. The Poly(A) Tail-Length Assay Kit (Thermo Fisher Scientific #76450) was then used to measure the poly(A) tail length of cyclin B1 mRNA. G/I tailing, reverse transcription, and PCR were performed according to the manual of the kit. Fragment 1 was obtained by PCR using a gene-specific forward primer (ccactcctgtcttgtaatgccac) and a gene-specific reverse primer (ccaccaataaatttattcaacttgatatcagatg)[39] located 8 nt upstream of the poly(A) start site. Fragment 2 was obtained by PCR using the same gene-specific forward primer and a 35 nt universal reverse primer (provided in the kit) that binds to the end of the cDNA. The PCR products were resolved on a 2.5% agarose gel. The poly(A) tail length was calculated as (PCR fragment 2) - (PCR fragment 1) − 8 − 35. To investigate whether acute depletion of PABPCs alters the poly(A) tail length of *Tcl1b4*, *Actb*, *Sec61g* and cyclin B1, gene-specific forward primers and the universal reverse primer were used for PCR. Forward primers: (cccaaccccaacttgtttttaactaattaaactttact) for *Tcl1b4*, (tacactgacttgagaccaataaaagtgcacac) for *Actb*, (tgttttgtttcttcctccttcc) for *Sec61g*, (ccactcctgtcttgtaatgccac) for cyclin B1.

## Image analysis and quantification

Images were analyzed and quantified in Fiji (NIH)[61], Imaris (Bitplane) or ZEN (Zeiss). The exported data were further processed in Excel and Graphpad Prism 9. Time-lapse movies of live oocytes were analyzed in both Imaris and ZEN. NEBD was defined as the time point when the sharp boundary between the nucleus and cytoplasm disappeared in the differential interference contrast (DIC) image. Anaphase onset was defined as the time point before chromosome separation was first observed. Chromosomes that failed to congress on the metaphase plate at anaphase onset were classified as misaligned chromosomes. The mean fluorescence intensity in oocytes was measured in Fiji or ZEN. Oocytes were manually outlined based on the fluorescent signal or the DIC images. The signal outside the oocyte in imaging dish was taken as background signal. To quantify the number of RIBOmap spots in Fiji, Watershed was applied to separate connected spots after thresholding. Projected images were quantified in all experiments, except for Supplementary Fig. 2f, where only the mid-section was quantified due to the very large number of spots in the whole oocyte. The data were normalized by dividing the values of each group by the mean of the control group in each experiment.

## Statistical analysis

The standard deviation (SD) was used as y-axis error bars for bar graphs plotted from the mean value of the data. The standard error of the mean (SEM) was used as y-axis error bars for curves plotted from the mean value of the data. Statistical significance based on unpaired two-tailed Student's t-test (for comparison of two groups) was calculated in Excel or Graphpad Prism 9. Statistical significance based on one-way/two-way ANOVA followed by Tukey's *post-hoc* test (for

comparison of three or more groups) or two-tailed Fisher's exact test was calculated in Graphpad Prism 9. All data are from at least two independent experiments. *P* values are designated as ****$P < 0.0001$, ***$P < 0.001$, **$P < 0.01$, *$P < 0.05$. Nonsignificant values are indicated as N.S.

## Reporting summary

Further information on research design is available in the Nature Portfolio Reporting Summary linked to this article.

## Data availability

The data that support the findings of this study are available from the corresponding author upon reasonable request. Source data are provided with this paper.

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

## Acknowledgements

We thank the staff of the animal facility and live-cell imaging facility of the Max Planck Institute for Multidisciplinary Sciences for technical assistance; M. Daniel and L. Wartosch for coordination of animal experiments; A. Heim and T.U. Mayer for helpful discussions. R. Singer for the plasmids phage-CMV-CFP-24×PP7. The research leading to these

results was funded by the Max Planck Society and Deutsche For-schungsgemeinschaft (DFG, German Research Foundation) under Germany's Excellence Strategy (EXC 2067/1-390729940) and a DFG Leibniz Prize to M.S. (SCHU 3047/1-1). Open Access funding provided by Max Planck Society.

## Author contributions

S.C. and M.S. conceived the study. S.C. and M.S. designed the experiments and methods for data analysis. S.C. performed all experiments and analyzed all data. S.C. and M.S. wrote the manuscript and prepared the figures. M.S. supervised the study and obtained funding.

## Funding

## Competing interests

The authors declare no competing interests.
