## [Transparent Peer Review file · Nature Communications]

Two mechanisms repress cyclin B1 translation to maintain prophase arrest in mouse oocytes

Corresponding Author: Professor Melina Schuh

Version 0:

Reviewer comments:

Reviewer #1

(Remarks to the Author)

Using the Trim-Away technique, the authors have identified two distinct mechanisms that regulate the translation of cyclin B1 during the prophase arrest of mouse oocytes. Firstly, the RNA-binding proteins DDX6 and LSM14B act to repress the translation of cyclin B1. Secondly, cytoplasmic poly(A)-binding proteins (PABPCs) indirectly repress the translation of cyclin B1 by sequestering the translation machinery on long-tailed mRNAs. This second pathway also plays a role in the translation of other mRNAs with either poly(A)-tail-less or short-tailed mRNAs during prophase arrest of mouse oocytes. These findings are quite interesting. However, the first pathway has been previously reported in several recent studies examining the function of LSM14B. The second pathway described in this study represents a significant finding for translation regulation during mouse oocyte maturation.

MAJOR ISSUES:

1. The authors have overstated the results of their study, which primarily focused on the translation of cyclin B1 during prophase arrest of mouse oocytes. Until they present comprehensive evidence to back up their claims regarding the translation regulation during this phase, the manuscript's title should undergo revision to reflect the actual scope of the research.
2. Due to the importance of PABPCs sequestering the translation machinery in this study, the proteomic analysis of PABPC1L or/and PABPC4 knockdown oocytes is strongly suggested. The proteomic analysis of DDX6 knockdown oocytes is also suggested.
3. MARDO protein ZAR1 and YBX2 should be examined in DDX6 knockdown oocytes.
4. From the results, it appears that DDX6 may specifically regulate Cyclin B1 mRNA. Any hit can be obtained from the result of LSM14B RIP from previous study (Li et al., *Advanced Science* 2023)? Otherwise, an RIP experiment of DDX6 to confirm its specific binding to mRNA is suggested.
5. It is preferred to use RIP-seq for DDX6 to further investigate its specific binding to mRNA. This approach will provide a comprehensive view of the mRNA targets that DDX6 binds to, which may aid in understanding DDX6 and LSM14B complex regulatory mechanism.
6. The cut-off point for PABP's regulation of poly(A) tail length is an interesting question, although it may not be easy to determine.

MINOR ISSUES

1. The authors should be careful to use terminology. For example, in lines 94-100, the definition of SN and NSN stages is not as clear as it used to be.
2. In Extended Data Fig. 1d, the antibody should be indicated to recognize the peptide region.
3. In Fig. 2c, the PLA experiment should include a MII as a positive control.
4. The results from RT-PCR indicate that there is no significant change in the mRNA levels of Cyclin B1. However, there is a noticeable difference in intensity of F2 in different samples in Fig. 3b.
5. In Reference 37, these authors only did the experiment from 0-6 hours. Thus, in lines 217-219, the description "reaching a maximum after 6 hours of maturation" may not be accurate. It is recommended to revise the description accordingly.
6. For the changes in poly(A) tail length shown in Fig. 4e, it is recommended to include corresponding PCR experiments to demonstrate the changes.
7. If Fig. 4i shows that individual knockdowns do not lead to meiotic resumption, how do they individually affect the translation of Cyclin B1?

The results in Fig. 1i show that co-expression of DDX6 and LSM14B can rescue meiotic resumption, while the results in Fig. 5 indicate that the binding of either DDX6 or LSM14B can effectively inhibit the translation of Cyclin B1. It is possible that the results in Fig. 1i are due to insufficient protein expression levels. It would be interesting to test whether increasing the concentration of injected DDX6/LSM14 mRNA could rescue meiotic resumption.

Reviewer #2

(Remarks to the Author)

In this paper Cheng and Schuh highlighted how repression of Cyclin B1 translation by a DDX6/LSM14B/4E-T complex is critical to maintain the prophase arrest. Indeed, acute depletion with the TRIM-Away of DDX6 results in precocious translation of cyclin B1 and spontaneous meiosis resumption in the presence of high PKA activity. The ectopic translation of cyclin B1 takes place in the absence of mRNA polyadenylation. These results suggest that two independent mechanisms can control the level of translation of cyclin B1: translation repression, independently of poly(A) length, and poly(A)-dependent translation activation, which occurs during meiosis resumption. They also propose an interesting and new model on how PABPCs participate in controlling translation through two mechanisms: first, by binding to long tailed mRNAs and activate directly their translation, as it was previously shown, and by repressing indirectly the translation of short tailed mRNAs. This second indirect mechanism is based on sequestering the translation machinery of the oocyte, which is a limiting factor for translation, on mRNAs with long poly(A) tails.

Impact and novelty:

Understanding the molecular mechanisms regulating translation repression and activation during the oocyte prophase arrest and meiosis resumption is very important to fully comprehend how meiotic cell divisions are regulated and how the oocyte becomes a fertilizable egg. The results presented in this paper are novel, impactful and well adapted to the large readership of Nature Communications. These important results include the demonstration that the acute depletion of either DDX6, CPEB1 or PABPC1L+PABPC4 induces spontaneous maturation. Furthermore, the model proposed on how PABPCs target the limiting translation machinery to polyadenylated mRNA to focus translation at their level is highly interesting, even though I have some questions on how this model fits some of the results (see comments on Fig.4F-G) and an important experiment is necessary to assess its validity (see comments on Fig.7B).

Rational and design of the experiments:

The experimental approaches used are appropriate and original. The experiments are overall well conducted with appropriate controls. However, an important control is missing about the stability of the length of poly(A) tail in the prophase-arrested oocytes (see comments on Fig. 3-4-5).

Fig. 2: The authors used an elegant approach to measure the increased association between CDK1 and cyclin B1 in oocytes where the level of cyclin B1 increases due to the acute depletion of DDX6. It is well established that the association of CDK1 with cyclin B1 should produce the formation of new CDK1/cyclin B1 complexes that should be phosphorylated at Y15, if CDK1 activation is maintained inhibited. Author should validate their elegant experimental approach measuring by western blot the increase in CDK1 phosphorylation at Y15 in oocyte depleted of DDX6.

The result that the acute depletion of CPEB1 in mouse oocytes induces spontaneous maturation is an important result and its visibility might be increased by moving it from the extended data into the main figure. The authors should also show that cyclin B1 level is increased upon CPEB1 depletion, either by western blot or by the modified RIBOmap assay that they developed.

Fig.4F-G: It is hard to fully fit some of these observations with the model of PABPCs proposed by the authors for two reasons. First, according to the model of the authors, in the absence of PABPCs, ribosomes should distribute equally on all mRNAs regardless of length of their poly(A) tail. This explains well the activation of translation observed with mClover-CcnB1UTR-A0 reporter which would be low in control conditions and increased in the absence of PABPCs. According to this model, in the absence of PABPCs, the translation of the mClover-CcnB1UTR-A0 reporter should occur at the same rate in prophase-arrested oocytes (Fig. 4F) and in oocytes resuming meiosis (Fig. 4G). This is because even though mClover-CcnB1UTR-A0 gets polyadenylated during meiosis resumption, its translation should not be sensitive to its polyadenylation or the polyadenylation of other endogenous mRNAs because of the absence of PABPCs. However, a difference in the accumulation of the mClover-CcnB1UTR-A0 reporter is observed between 4F and 4G both at 6h and 12h in the absence of PABPCs, which contradicts this model.

Second, according to the model of the authors, PABPCs role is to sense the length of the poly(A) tail and target the limited translation machinery to focus only on mRNAs with longer poly(A) tail. During meiosis resumption, the mClover-CcnB1UTR-A0 reporter gets extensively polyadenylated. Hence, in control conditions, PABPCs should sense this extension and increase extensively the translation of the reporter. Therefore, the translation of mClover-CcnB1UTR-A0 reporter in this condition should be much higher than the translation resulting from the absence of PABPCs, which is due to the dilution of the limiting translation machinery among all the mRNAs. This is not the case in Fig. 4G.

Potentially, better measurements of the translation rates in the 4 different conditions should be obtained by analyzing the translation of the reporter with a higher temporal resolution, which is possible with this time-lapse based approach on alive oocytes.

Figs. 3-4-5: In Figs. 3-4-5, the authors inject various translation reporters with different lengths of poly(A) tail. The basic assumption of all these experiments is that the reporters will maintain the same length of poly(A) tails in prophase-arrested oocytes during the course of the experiment. This assumption might be incorrect since adenylase/deadenylase activities might be present in the oocytes, even during the prophase arrest, which could act elongating/reducing the length of the poly(A) of the reporters. This has been already suggested in mouse oocytes (10.1093/nar/gkaa010). In this paper, the same

translation reporter was produced with different lengths of the poly(A) tail and they were injected in the oocytes arrested in prophase. The rate of translation of these reporters at the beginning was different, because of the different lengths of poly(A) tail, but with time their rate of translation converged to the same level, suggesting that the length of their poly(A) tail has changed. The level of translation to which they were converging depends on the information (or lack of information) present in the 3'UTR.

I would suggest to use a PAT-assay (as the authors have done for endogenous cyclin B1 in Fig. 3B) to measure the length of the poly(A) tails of the reporters (A0, A34, and A>150) to see if it changes after 6h and 12h of injection in oocytes that are maintained arrested in prophase. This should be done for both the mClover3 reporter and the mClover3-Ccnb1 3'UTR reporters, since the information in the 3'UTRs or the lack of UTR could result in a different equilibrium of adenylase/deadenylase activities on the reporter. The stability of the reporter should be measured by qPCR to assess its stability since the absence of poly(A) could result in a partial instability of the reporter.

This control is very important also for Fig. 5 in which the length of poly(A) tail is used to estimate the number of PABPCs bound to the reporter. These experiments are important to validate the model for translation summarized in Fig. 7.

Fig. 7B: The authors proposed that in the absence of PABPCs, ribosomes are distributed at the same extent on the mRNAs regardless of the length of the poly(A) tail. This has been shown for mClover3 reporter without UTR in Fig. 4A-B-C, and for mClover-CcnB1UTR in Fig. 4E-F-G. However, comparing the same reporter with different poly(A) tails does not allow to evaluate the contribution of the information of the 3'UTR, which also plays a role in determining the level of translation of the reporter, and therefore the ribosome recruitment.

I would suggest the author to compare the translation of mClover-CcnB1UTR and mScarlet-noUTR (both reporters without poly(A) tail) after co-injection in prophase oocytes where PABPCs are acutely depleted. Such an experiment would indicate if the elements present in 3'UTR can still produce a differential recruitment of the ribosomes on the reporter, or if they are not anymore influent when the poly(A) sensing machinery is removed.

Version 1:

Reviewer comments:

Reviewer #1

(Remarks to the Author)

The authors have fully addressed my concerns.

Reviewer #2

(Remarks to the Author)

I have reviewed the revised manuscript along with the authors' responses to my comments and suggestions. I really appreciate the authors for their efforts in revising the manuscript, particularly the addition of the new experiments in new Fig. 4 and the improved presentation of results in new Fig. 5.

The results presented in this revised article are novel, relevant, and impactful. I believe this significant contribution will be well received by the broad readership of Nature Communications and I support its publication.

Response to referees ID: NCOMMS-24-03743-T

Two mechanisms repress cyclin B1 translation to maintain prophase arrest in mouse oocytes

First, we want to thank both referees for their insightful comments and expert suggestions that greatly improved this study.

REVIEWER COMMENTS

Reviewer #1 (Remarks to the Author):

Using the Trim-Away technique, the authors have identified two distinct mechanisms that regulate the translation of cyclin B1 during the prophase arrest of mouse oocytes. Firstly, the RNA-binding proteins DDX6 and LSM14B act to repress the translation of cyclin B1. Secondly, cytoplasmic poly(A)-binding proteins (PABPCs) indirectly repress the translation of cyclin B1 by sequestering the translation machinery on long-tailed mRNAs. This second pathway also plays a role in the translation of other mRNAs with either poly(A)-tail-less or short-tailed mRNAs during prophase arrest of mouse oocytes. These findings are quite interesting. However, the first pathway has been previously reported in several recent studies examining the function of LSM14B. The second pathway described in this study represents a significant finding for translation regulation during mouse oocyte maturation.

First, we would like to thank the reviewer for the positive feedback and expert suggestions on our study. We have addressed the comments as detailed below, and feel that the changes and experimental additions have greatly improved the manuscript.

We would like to clarify why the recently published papers on LSM14B (Li et al., Adv Sci 2023; Zhang et al., J Genet Genomics 2023; Shan et al., Cell Mol Life Sci 2023; Wan et al., Nucleic Acids Res) do not affect the novelty of the first part of our study.

First, our work complements but does not overlap with these papers on the LSM14B aspect. These papers used *Lsm14b* knockout mice to deplete LSM14B in oocytes, whereas we used Trim-Away to acutely deplete DDX6 and LSM14B only in late-stage oocytes. We found that acute depletion of DDX6/LSM14B impaired oocyte prophase arrest, which was not reported in any previous papers (**previous Fig. 1 or new Fig. 1**). *Lsm14b* gene knockout also affects early oogenesis (Shan et al., Cell Mol Life Sci 2023). This may be the reason why *Lsm14b* gene knockout and LSM14B Trim-Away had some different effects on late-stage oocytes.

Second, by combining Trim-Away and rescue experiments, we found that both DDX6 and LSM14B are required for maintaining prophase arrest (**previous Fig. 1 or new Fig. 1**).

Third, we performed a series of experiments (**previous Fig. 2 and previous Extended Data Fig. 2 or new Fig. 2 and new Extended Data Fig. 2**) to discover and validate that

DDX6/LSM14B represses cyclin B1 translation to maintain prophase arrest. In particular, we modified the recently developed RIBOmap assay (Zeng et al., Science 2023) and successfully performed it in mouse oocytes using a simpler detection method (**previous Fig. 2e,f or new Fig. 2d,e**). This modified RIBOmap assay provided direct evidence that DDX6/LSM14B represses cyclin B1 translation (**previous Fig. 2g,h or new Fig. 2f,g**). We also performed cyclin B1 overexpression and RNAi to confirm that premature cyclin B1 translation is the cause of premature meiotic resumption upon acute depletion of DDX6/LSM14B (**previous Fig. 2j,l or new Fig. 2h,i**). Furthermore, in the revised manuscript, our new data indicate that the specificity of DDX6/LSM14B for cyclin B1 and other mRNAs is achieved by the CPEB1 protein and the cytoplasmic polyadenylation elements (CPEs) in the 3'UTR of these mRNAs (**new Fig. 2k and new Extended Data Fig. 2e,f**). Together, we have shown mechanistically that DDX6/LSM14B and CPEB1 repress cyclin B1 translation to maintain prophase arrest, which was not reported in any previous studies.

Thus, our study and the previously published work on LSM14B complement each other and are both important for a better understanding of the functions of LSM14B.

MAJOR ISSUES:

1. The authors have overstated the results of their study, which primarily focused on the translation of cyclin B1 during prophase arrest of mouse oocytes. Until they present comprehensive evidence to back up their claims regarding the translation regulation during this phase, the manuscript's title should undergo revision to reflect the actual scope of the research.

We thank the reviewer for highlighting this point. We have had some difficulty in choosing a concise title for this paper as several proteins were studied. Following the reviewer's suggestion, we have now changed the title to "Two mechanisms repress cyclin B1 translation to maintain prophase arrest in mouse oocytes".

2. Due to the importance of PABPCs sequestering the translation machinery in this study, the proteomic analysis of PABPC1L or/and PABPC4 knockdown oocytes is strongly suggested. The proteomic analysis of DDX6 knockdown oocytes is also suggested.

We thank the reviewer for these helpful suggestions. We agree that proteomic analyses could complement our study. However, carrying out these experiments is not possible for technical reasons. Mass spectrometry requires large numbers of mouse oocytes (from hundreds to thousands) to obtain reliable data with a good coverage rate. Trim-Away was performed through quantitatively injecting oocytes one by one, which makes it impossible to obtain large numbers of oocytes. In addition, based on our experience with mouse oocyte mass spectrometry, only the most abundant proteins (many of which are housekeeping proteins) can be accurately measured when using at least 3 x 100 oocytes. Unfortunately, a Trim-Away study of > 300 oocytes for one sample cannot be conducted.

We would also like to highlight that our study focuses on understanding the mechanism behind the unique protracted prophase arrest in oocytes. Omics-based experiments would have been helpful if we had wanted to identify all proteins that are regulated by poly(A)-binding proteins (PABPCs) or DDX6/LSM14B. But this is not the aim of our study and we fear that adding such experiments would distract from the main storyline. Therefore, having observed that DDX6 Trim-Away impaired prophase arrest (**previous Fig. 1 or new Fig. 1**), we naturally focused on cell cycle regulators. Indeed, through a candidate-based screen on cyclin B1, MOS and CDK1 and further characterization experiments (**previous Fig. 2 and previous Extended Data Fig. 2 or new Fig. 2 and new Extended Data Fig. 2**), we have validated that the premature translation of cyclin B1 is the main cause of premature meiotic resumption. In the revised manuscript, we have also excluded the possibility that post-translational modification of CDK1 is changed in the first 6 hours after DDX6 Trim-Away (**new Fig. 2c and new Extended Data Fig. 2a**). In this context, the use of an omics-based screen to find possible targets of DDX6 or PABPCs is no longer necessary. By performing proteomic analysis, we may be able to find some differentially expressed proteins, but many more validation experiments would be needed to confirm whether DDX6 or PABPCs truly regulate their translation, and these data would not be related to prophase arrest. We therefore consider the proteomic analysis to be beyond the scope of our study and a distraction from the main storyline.

Given these limitations, we addressed the reviewer's concern in an alternative way. The main concern of the reviewer appears to be whether the mechanism by which PABPCs regulate translation is applicable to other mRNAs. Using the modified RIBOmap assay, we have shown that PABPCs also regulate the translation of other mRNAs with short poly(A) tails in the same way that we proposed. To further address this point, in the revised manuscript, we have unbiasedly selected a few published mRNAs with different poly(A) tail lengths based on some reasonable criteria (**new Extended Data Fig. 4c**), and performed the modified RIBOmap assay to directly test how acute depletion of PABPCs affects their translation. As expected, PABPC Trim-Away increased the translation of short-tailed mRNAs and decreased the translation of long-tailed mRNAs (**new Fig. 4e-i**). By contrast, the translation of *Actb* mRNA, the tail length of which is intermediate between the other two groups, was not significantly changed (**new Fig. 4e-i**). Moreover, acute depletion of PABPCs did not change the stability or the poly(A) tail length of these mRNAs (**new Extended Data Fig. 4d-l**). These data correlate well with the reporter assay, in which the only variant is the length of the poly(A) tail (**new Fig. 4a-d**). Taken together, the mechanism by which PABPCs regulate translation is applicable to other mRNAs. It is very likely that the poly(A) tail length of an mRNA is the only factor that determines how PABPCs regulate its translation. These new data are now described on page 8 line 211-222 of the revised manuscript.

3. MARDO protein ZAR1 and YBX2 should be examined in DDX6 knockdown oocytes.

We thank the reviewer for this helpful suggestion. We have now added these data in **new Extended Data Fig. 1c,d**. The protein levels of ZAR1 and YBX2 were not significantly changed in DDX6 knockdown oocytes. These new data are now described on page 3 line 85-86 of the revised manuscript.

4. From the results, it appears that DDX6 may specifically regulate Cyclin B1 mRNA. Any hit can be obtained from the result of LSM14B RIP from previous study (Li et al., Advanced Science 2023)? Otherwise, an RIP experiment of DDX6 to confirm its specific binding to mRNA is suggested.

We thank the reviewer for these helpful suggestions.

We agree with the reviewer that it is a very interesting question whether DDX6/LSM14B regulates only cyclin B1 or a group of mRNAs. We have now addressed this point in the revised manuscript. Please see the details in the following point.

5. It is preferred to use RIP-seq for DDX6 to further investigate its specific binding to mRNA. This approach will provide a comprehensive view of the mRNA targets that DDX6 binds to, which may aid in understanding DDX6 and LSM14B complex regulatory mechanism.

The paper the reviewer refers to (Li et al., Adv Sci 2023) used LACE-Seq to identify LSM14B-bound mRNAs and obtained 9253 target mRNAs. The large number of target mRNAs (equivalent to almost half of protein-coding genes) suggests that LSM14B might be a universal RNA-binding protein. As an RNA-helicase, DDX6 also appears to be a universal RNA-binding protein. Thus, RIP-seq (or other similar methods) for DDX6 or LSM14B would identify very large numbers of mRNAs and would not allow us to identify functionally relevant subgroups of mRNAs regulated by DDX6/LSM14B.

Given these circumstances, we aimed to identify the mRNA targets regulated by DDX6/LSM14B in an alternative way. In the first version of our manuscript, we have shown that partial depletion of CPEB1 by Trim-Away also caused premature meiotic resumption, similar to acute depletion of DDX6/LSM14B (**previous Extended Data Fig. 2b,c or new Fig. 2j,l**). Co-IP experiments indicated that CPEB1 is in a complex with DDX6/LSM14B (Li et al., Adv Sci 2023; Minshall et al., J Biol Chem 2007). CPEB1 is known to be able to bind a subset of mRNAs containing CPEs in the 3'UTR, including cyclin B1 (Luong et al., Nucleic Acids Res 2020). Thus, it is likely that CPEB1 confers specificity of DDX6/LSM14B to a subset of mRNAs.

To further validate this point, we have included new data in the revised manuscript. We found that partial depletion of CPEB1 by Trim-Away increased cyclin B1 translation (**new Fig. 2j,k**), which is also similar to depletion of DDX6/LSM14B (**new Fig. 2f,g or previous Fig. 2g,h**). The phenotype of CPEB1 Trim-Away is weaker because CPEB1 proteins were only partially depleted. Together with our previous data referred to above, it is sufficient to conclude that DDX6/LSM14B and CPEB1 act together to repress cyclin B1 translation and maintain prophase arrest. To test whether this mechanism is applicable to other CPEB1-bound mRNAs, we selected a few validated CPEB1-bound mRNAs containing multiple CPEs from a published paper (Luong et al., Nucleic Acids Res 2020) and performed the modified RIBOmap assay. We found that acute depletion of DDX6/LSM14B increased the translation of mRNAs containing CPEs, while having no obvious effect on mRNAs without CPEs (**new Extended Data Fig. 2e,f**). Thus, the mechanism by which DDX6/LSM14B and CPEB1 regulate translation is

applicable to other CPEB1-bound mRNAs. These new data are now described on page 6 line 146-153 of the revised manuscript.

By combining Trim-Away, rescue experiments, Western blot, qPCR, modified RIBOmap, overexpression and RNAi, our study has now mechanistically revealed how DDX6/LSM14B and CPEB1 repress cyclin B1 translation to maintain prophase arrest and also suggests that this mechanism is applicable to other CPEB1-bound mRNAs.

6. The cut-off point for PABP's regulation of poly(A) tail length is an interesting question, although it may not be easy to determine.

We thank the reviewer for highlighting this interesting point. In the revised manuscript, we have now shown that acute depletion of PABPCs increased the translation of short-tailed endogenous mRNAs and decreased the translation of long-tailed endogenous mRNAs, while not significantly changing the translation of *Actb*, which has a ~70 nt poly(A) tail (**new Fig. 4e-i and new Extended Data Fig. 4c-l**). Thus, it is possible that the cut-off point is close to 70 nt. We now discuss this point on page 8 line 211-222.

MINOR ISSUES

1. The authors should be careful to use terminology. For example, in lines 94-100, the definition of SN and NSN stages is not as clear as it used to be.

We thank the reviewer for highlighting this point. We have now revised the text in the manuscript to address this point.

2. In Extended Data Fig. 1d, the antibody should be indicated to recognize the peptide region.

The DDX6 antibody we used for Trim-Away is a commercial antibody whose corresponding antigen has not been mapped. Therefore, we could not design point mutations to perform the rescue experiments. To overcome this problem, we came up with the idea of using a DDX6 homologue to perform the rescue experiments. CGH-1, the *C. elegans* DDX6, is highly similar to the mouse DDX6, but also has many amino acid variations (**new Extended Data Fig. 1f or previous Extended Data Fig. 1d**). The antigen that the antibody recognizes must therefore be located in a region that differs between *C. elegans* and mouse DDX6. We now highlight this point on page 31 line 930-933 of the revised manuscript.

3. In Fig. 2c, the PLA experiment should include a MII as a positive control.

We thank the reviewer for this helpful suggestion.

We have now performed this experiment. Surprisingly, we found that the number of CDK1-cyclin B1 complexes detected by PLA was not significantly increased in MII oocytes compared to GV oocytes (**Response Fig. 1a**). Next, we compared maturing oocytes (5 hours) with GV oocytes, and again we did not see any obvious change (**Response Fig. 1b**). However, when we compared prophase-arrested NSN oocytes and SN oocytes, an obvious increase in the CDK1-cyclin B1 complex was observed during the NSN to SN transition (**Response Fig. 1c,d**). This is consistent with a previous study showing that CDK1 and cyclin B1 proteins increase during oocyte growth (Kanatsu-Shinohara et al., Biol Reprod 2000), and therefore suggests that the PLA assay is able to detect the CDK1-cyclin B1 complex.

These data indicate that the regulation of CDK1-cyclin B1 complex formation during the prophase to MII transition is more complicated than previously thought, perhaps because of a change in CDK1 phosphorylation state or the expression of other cyclins or CDKs. Nevertheless, the regulation of CDK1-cyclin B1 complex formation in prophase-arrested oocytes, either in the case of NSN to SN transition or in the case of DDX6 Trim-Away in the presence of RO-3306, is more straightforward.

As the PLA data in maturing oocytes is surprising and beyond current knowledge, we have decided to remove the PLA data to avoid any potential confusion. Removing the PLA data does not affect any of the main conclusions of our study, as we have plenty of data (Trim-Away, rescue experiments, Western blot, qPCR, modified RIBOmap, overexpression and RNAi) to support that DDX6/LSM14B represses cyclin B1 translation to maintain prophase arrest.

Response Fig. 1: CDK1-cyclin B1 complexes detected by PLA increase during NSN to SN transition. **a**, CDK1-cyclin B1 complexes detected by PLA do not change significantly during GV to MII transition. **b**, CDK1-cyclin B1 complexes detected by PLA do not change significantly before (GV) or after (5 h) meiotic resumption. **c,d**, CDK1-cyclin B1 complexes detected by PLA are significantly increased during NSN to SN transition. The number of analyzed oocytes is specified in italics. Data are shown as mean \pm SD. *P* values were calculated using unpaired two-tailed Student's *t* test. Scale bar, 10 μ m.

4. The results from RT-PCR indicate that there is no significant change in the mRNA levels of Cyclin B1. However, there is a noticeable difference in intensity of F2 in different samples in Fig. 3b.

We thank the reviewer for highlighting this point. RT-qPCR is a suitable method for comparing mRNA levels, but the Poly(A) Tail-Length Assay (PAT assay) is not quantitative. In the last step of the PAT assay, a PCR of 35 cycles was performed, which is perhaps beyond the exponential period and therefore cannot accurately reflect the template level. It is worth noting that performing the PAT assay using very limited material is very challenging, so it is very common for the PCR band to be faint or smeared. The main purpose of the PAT assay is to assess whether the poly(A) tail length of the main population of an mRNA is altered or not.

5. In Reference 37, these authors only did the experiment from 0-6 hours. Thus, in lines 217-219, the description "reaching a maximum after 6 hours of maturation" may not be accurate. It is recommended to revise the description accordingly.

We thank the reviewer for highlighting this point. Following the reviewer's suggestion, we have now deleted the words "reaching a maximum after 6 hours of maturation".

6. For the changes in poly(A) tail length shown in Fig. 4e, it is recommended to include corresponding PCR experiments to demonstrate the changes.

We thank the reviewer for this helpful suggestion. As shown in **previous Fig. 3a (new Fig. 3a)**, the PCR step of the PAT assay requires a forward primer that recognizes the 3'UTR. However, this primer cannot distinguish between endogenous cyclin B1 mRNA and the *mClover3-Ccnb1* 3'UTR reporter mRNA. To specifically detect the reporter mRNA, we need to use a primer that recognizes *mClover3* (**Response Fig. 2a**). This means that the PCR band will be larger than 1 kb. As mentioned in Minor Point 4, using very limited materials (oocytes) to perform the PAT assay is very challenging. It is almost impossible to amplify a fragment larger than 1 kb in this assay. Even if successful, this fragment is also too large to reflect a small change in poly(A). We have now performed this assay. Consistent with the explanations above, only smeared signals were observed (**Response Fig. 2b**).

It is widely accepted that the poly(A) tails of cyclin B1 and many other CPEB1-bound mRNAs are elongated when the oocyte resumes meiosis. Reporter assays are also widely used to show the increase in translation upon poly(A) tail elongation (Yu et al., Nat Struct Mol Biol 2016; Sha et al., Development 2017; Yang et al., Genes Dev 2017; Dai et al., Nucleic Acids Res 2019; Luong et al., Nucleic Acids Res 2020; Jiang et al., Nucleic Acids Res 2021). Furthermore, the increase in translation of the reporter mRNAs is dependent on the phosphorylation of the poly(A) polymerase α (Jiang et al., Nucleic Acids Res 2021). We hope the reviewer agrees that the PAT assay is not necessary here.

Response Fig. 2: Measurement of the poly(A) tail length of the *mClover3-Ccnb1 3'UTR* reporter mRNA is not possible. a, Strategy for measuring the poly(A) tail length of the reporter mRNA. **b**, The expected PCR product is too large for a successful PAT assay.

7. If Fig. 4i shows that individual knockdowns do not lead to meiotic resumption, how do they individually affect the translation of Cyclin B1?

We thank the reviewer for this helpful suggestion. Following the reviewer's suggestion, we have now performed the modified RIBOmap assay to test whether acute depletion of PABPC1L or PABPC4 alone affects cyclin B1 translation. As shown in **new Fig. 5a-c**, acute depletion of PABPC1L slightly increased cyclin B1 translation, but this was not sufficient to cause premature meiotic resumption. Acute depletion of PABPC4 had no obvious effect on cyclin B1 translation and prophase arrest. By contrast, co-depletion of PABPC1L and PABPC4 significantly increased cyclin B1 translation and therefore caused premature meiotic resumption. These data demonstrate the redundancy of PABPC1L and PABPC4. These new data are now described on page 9 line 234-239 of the revised manuscript.

The results in Fig. 1i show that co-expression of DDX6 and LSM14B can rescue meiotic resumption, while the results in Fig. 5 indicate that the binding of either DDX6 or LSM14B can effectively inhibit the translation of Cyclin B1. It is possible that the results in Fig. 1i are due to insufficient protein expression levels. It would be interesting to test whether increasing the concentration of injected DDX6/LSM14 mRNA could rescue meiotic resumption.

We thank the reviewer for highlighting this interesting point.

In **previous Fig. 1i (new Fig. 1i)**, CGH-1-mClover3 (*C. elegans* DDX6) and LSM14B-mScarlet have already been overexpressed. In **previous Fig. 5 (new Fig. 6)**, it appears that PP7-tdPCP-mediated binding of either DDX6 or LSM14B is sufficient to effectively inhibit translation of the bound mRNA, but we cannot exclude the potential role of endogenous DDX6/LSM14B in this assay. We hypothesized that tdPCP-DDX6 requires endogenous LSM14B to effectively repress translation of the bound mRNA and vice versa. To test this hypothesis, we used tdPCP-CGH-1 (*C. elegans* DDX6) to repress translation of the bound mRNA (**Response Fig. 3**). We found that when endogenous DDX6/LSM14B was depleted by Trim-Away, the repression effect of tdPCP-CGH-1 on the bound mRNA was significantly

attenuated. CGH-1 can complement endogenous DDX6. Therefore, loss of endogenous LSM14B impaired the repressive effect of tdPCP-CGH-1. In the future, we can potentially use a similar assay to test other components that are required for forming the repression complex, such as 4E-T, but these are beyond the scope of the current study.

Response Fig. 3: Translational repression of CGH-1 (*C. elegans* DDX6) on the bound mRNA depends on LSM14B. **a,b**, tdPCP-CGH-1 efficiently represses the translation of the bound mRNA, which is attenuated when endogenous DDX6/LSM14B is depleted. The number of analyzed oocytes is specified in italics. Data are shown as mean \pm SD. *P* values were calculated using one-way ANOVA with Tukey's *post-hoc* test. Scale bar, 10 μ m.

Reviewer #2 (Remarks to the Author):

In this paper Cheng and Schuh highlighted how repression of Cyclin B1 translation by a DDX6/LSM14B/4E-T complex is critical to maintain the prophase arrest. Indeed, acute depletion with the TRIM-Away of DDX6 results in precocious translation of cyclin B1 and spontaneous meiosis resumption in the presence of high PKA activity. The ectopic translation of cyclin B1 takes place in the absence of mRNA polyadenylation. These results suggest that two independent mechanisms can control the level of translation of cyclin B1: translation repression, independently of poly(A) length, and poly(A)-dependent translation activation, which occurs during meiosis resumption. They also propose an interesting and new model on how PABPCs participate in controlling translation through two mechanisms: first, by binding to long tailed mRNAs and activate directly their translation, as it was previously shown, and by repressing indirectly the translation of short tailed mRNAs. This second indirect mechanism

is based on sequestering the translation machinery of the oocyte, which is a limiting factor for translation, on mRNAs with long poly(A) tails.

We thank the reviewer for the very nice summary of our study.

Impact and novelty:

Understanding the molecular mechanisms regulating translation repression and activation during the oocyte prophase arrest and meiosis resumption is very important to fully comprehend how meiotic cell divisions are regulated and how the oocyte becomes a fertilizable egg. The results presented in this paper are novel, impactful and well adapted to the large readership of Nature Communications. These important results include the demonstration that the acute depletion of either DDX6, CPEB1 or PABPC1L+PABPC4 induces spontaneous maturation. Furthermore, the model proposed on how PABPCs target the limiting translation machinery to polyadenylated mRNA to focus translation at their level is highly interesting, even though I have some questions on how this model fits some of the results (see comments on Fig.4F-G) and an important experiment is necessary to assess its validity (see comments on Fig.7B).

We thank the reviewer for the very positive comments on our study. We have now performed new experiments to address the concerns of the reviewer. The details are shown below. We believe that the changes and experimental additions have greatly improved the manuscript.

Rational and design of the experiments:

The experimental approaches used are appropriate and original. The experiments are overall well conducted with appropriate controls. However, an important control is missing about the stability of the length of poly(A) tail in the prophase-arrested oocytes (see comments on Fig. 3-4-5).

We thank the reviewer for the positive feedback on our study. We have now performed new experiments to address the reviewer's concerns. The details are shown below.

Fig. 2: The authors used an elegant approach to measure the increased association between CDK1 and cyclin B1 in oocytes where the level of cyclin B1 increases due to the acute depletion of DDX6. It is well established that the association of CDK1 with cyclin B1 should produce the formation of new CDK1/cyclin B1 complexes that should be phosphorylated at Y15, if CDK1 activation is maintained inhibited. Author should validate their elegant experimental approach measuring by western blot the increase in CDK1 phosphorylation at Y15 in oocyte depleted of DDX6.

We thank the reviewer for this helpful suggestion.

In the revised manuscript, we first verified the anti-CDK1(Y15p) antibody by staining prophase-arrested oocytes and maturing oocytes. CDK1 was phosphorylated at Y15 in prophase and dephosphorylated when the oocytes resumed meiosis (**new Extended Data Fig. 2a**), suggesting that this antibody works well. However, when we compared the CDK1(Y15p) levels in control oocytes and oocytes depleted of DDX6, no obvious difference was seen (**new Fig. 2c**). It is possible that the majority of CDK1 is phosphorylated in prophase-arrested oocytes, as partial replacement of endogenous CDK1 with phospho-dead CDK1 is sufficient to cause premature meiotic resumption (Adhikari et al., Cell Res 2016). We have previously shown that acute depletion of DDX6 did not change total CDK1 protein levels (**previous Fig. 2b or new Fig. 2b**). If total CDK1 levels are not increased, it is unlikely that CDK1(Y15p) would be increased, as the majority of CDK1 (if not all) is already phosphorylated. Although these data cannot be used to validate the PLA assay, they indicate that acute depletion of DDX6 does not change the phosphorylation state of CDK1 in the first 6 hours. Taken together with the Western blot of cyclin B1, MOS and total CDK1 (**previous Fig. 2a,b or new Fig. 2a,b**), these experiments indicate that the premature translation of cyclin B1 is the earliest event upon acute depletion of DDX6. These new data are now described on page 5 line 122-127 of the revised manuscript.

The first reviewer suggested that we use PLA to compare GV and MII oocytes. Surprisingly, we did not see an obvious increase in CDK1-cyclin B1 complex levels detected by PLA during the prophase to MII transition (**Response Fig. 1a**). However, using the PLA assay, we saw an obvious increase in CDK1-cyclin B1 complex during the NSN to SN transition (**Response Fig. 1c,d**). This is consistent with a previous work showing that CDK1 and cyclin B1 proteins increase during oocyte growth (Kanatsu-Shinohara et al., Biol Reprod 2000). The data in NSN and SN oocytes suggests that PLA is able to detect the CDK1-cyclin B1 complex. It appears that the regulation of CDK1-cyclin B1 complex formation in prophase-arrested oocytes, either in the case of NSN to SN transition or in the case of DDX6 Trim-Away in the presence of RO-3306, is very straightforward. By contrast, the regulation of CDK1-cyclin B1 complex formation during the prophase to MII transition is more complicated than previously thought, perhaps because of a change in CDK1 phosphorylation state or the expression of other cyclins or CDKs. As the PLA result in MII oocytes is surprising and beyond current knowledge, we have decided to remove the PLA data from our manuscript to avoid any potential confusion. Removing the PLA data does not affect any of the main conclusions of our study, as we have plenty of data (Trim-Away, rescue experiments, Western blot, qPCR, modified RIBOmap, overexpression and RNAi) to support that DDX6/LSM14B represses cyclin B1 translation to maintain prophase arrest. The regulation of CDK1-cyclin B1 complex formation during oocyte maturation needs to be investigated in more detail, but we feel that this is beyond the scope of the current study.

The result that the acute depletion of CPEB1 in mouse oocytes induces spontaneous maturation is an important result and its visibility might be increased by moving it from the extended data into the main figure. The authors should also show that cyclin B1 level is increased upon CPEB1 depletion, either by western blot or by the modified RIBOmap assay that they developed.

We thank the reviewer for these helpful suggestions.

Following the reviewer's suggestion, we have now moved all the CPEB1 data from the extended data to the main figure (**new Fig. 2j,l**). We have now also performed the modified RIBOmap assay to show that the acute CPEB1 depletion also causes increased translation of cyclin B1 (**new Fig. 2k**). As we can only partially deplete CPEB1 by Trim-Away, the phenotype is weaker than for DDX6 Trim-Away. These new data are now described on page 6 line 146-153 of the revised manuscript.

Fig.4F-G: It is hard to fully fit some of these observations with the model of PABPCs proposed by the authors for two reasons. First, according to the model of the authors, in the absence of PABPCs, ribosomes should distribute equally on all mRNAs regardless of length of their poly(A) tail. This explains well the activation of translation observed with mClover-CcnB1UTR-A0 reporter which would be low in control conditions and increased in the absence of PABPCs. According to this model, in the absence of PABPCs, the translation of the mClover-CcnB1UTR-A0 reporter should occur at the same rate in prophase-arrested oocytes (Fig. 4F) and in oocytes resuming meiosis (Fig. 4G). This is because even though mClover-CcnB1UTR-A0 gets polyadenylated during meiosis resumption, its translation should not be sensitive to its polyadenylation or the polyadenylation of other endogenous mRNAs because of the absence of PABPCs. However, a difference in the accumulation of the mClover-CcnB1UTR-A0 reporter is observed between 4F and 4G both at 6h and 12h in the absence of PABPCs, which contradicts this model.

We thank the reviewer for highlighting this interesting point. We agree with the reviewer that in the absence of PABPCs, the translation of the *mClover3-CcnB1 3'UTR* reporter mRNA should occur at the same (or similar) rate whether its poly(A) tail is elongated or not. However, in addition to the change in poly(A) tail length, there is another variable here: the stage of the oocyte. In the absence of PABPCs, the difference in translation of the *mClover-CcnB1 3'UTR* reporter mRNA in prophase-arrested oocytes and in oocytes resuming meiosis (**previous Fig. 4e,f,g or new Fig. 5e,f,h**) may be due to the fact that these oocytes are at different developmental stages.

To test this hypothesis, we expressed mRNAs with different poly(A) tail lengths in both prophase-arrested oocytes and oocytes resuming meiosis (**Response Fig. 4**). For all types of mRNAs, translation levels were always higher in prophase-arrested oocytes (RO-3306) than in oocytes resuming meiosis (DMSO) (**Response Fig. 4**). This is consistent with the observation that the *mClover-CcnB1 3'UTR* reporter was translated higher in prophase-arrested oocytes (RO-3306) than in oocytes resuming meiosis (DMSO) when PABPCs were acutely depleted (**previous Fig. 4e,f,g or new Fig. 5e,f,h**). Furthermore, we found that the difference in translation level was not due to the change in mRNA stability at different developmental stages (**new Extended Data Fig. 5j**). Taken together, reporter mRNAs are generally translated more efficiently in prophase-arrested oocytes than in maturing oocytes. This is a very interesting finding, and could be investigated further in the future.

Response Fig. 4: reporter mRNAs are generally translated more efficiently in prophase-arrested oocytes than in maturing oocytes. Data are shown as mean \pm SEM. P values were calculated using one-way ANOVA with Tukey's *post-hoc* test. Scale bar, 10 μ m.

Second, according to the model of the authors, PABPCs role is to sense the length of the poly(A) tail and target the limited translation machinery to focus only on mRNAs with longer poly(A) tail. During meiosis resumption, the mClover-CcnBIUTR-A0 reporter gets extensively polyadenylated. Hence, in control conditions, PABPCs should sense this extension and increase extensively the translation of the reporter. Therefore, the translation of mClover-CcnBIUTR-A0 reporter in this condition should be much higher than the translation resulting from the absence of PABPCs, which is due to the dilution of the limiting translation machinery among all the mRNAs. This is not the case in Fig. 4G.

Potentially, better measurements of the translation rates in the 4 different conditions should be obtained by analyzing the translation of the reporter with a higher temporal resolution, which is possible with this time-lapse based approach on alive oocytes.

We thank the reviewer for this helpful suggestion.

We agree with the reviewer that after extensive polyadenylation of the *mClover3-Ccnb1-3'UTR* reporter mRNA, its translation should be much higher in control oocytes than in PABPC-depleted oocytes. This is also what we have observed (**previous Fig. 4g,h or new Fig. 5h,i**). In the **previous Fig. 4g (new Fig. 5h)**, we quantified the fluorescence intensity at different time points. During the first 6 hours of oocyte maturation, when the poly(A) tail of the reporter mRNA is not long enough, its expression is higher in PABPC-depleted oocytes. This is consistent with our model that PABPC depletion increases the translation of short-tailed mRNAs. By contrast, in the next 6 hours of oocyte maturation, when the poly(A) tail of the reporter mRNA is already extensively elongated, its expression is now faster in control oocytes

than in PABPC-depleted oocytes. This is also consistent with our model that PABPC depletion reduces the translation of long-tailed mRNAs. Although the expression of the reporter mRNA is higher during the second 6 hours in control oocytes, its expression is lower during the first 6 hours. Therefore, we did not see an obvious increase in fluorescence intensity in control oocytes at 12 h. To avoid confusion, we also quantified the change in fluorescence intensity from 0 h to 6 h and from 6 h to 12 h in the **previous Fig. 4h** (**new Fig. 5i**).

We thank the reviewers for suggesting a more intuitive way of presenting the data. Following the reviewer's suggestion, we have now quantified the translation rate at a higher temporal resolution (**new Fig. 5g,j**). In prophase arrested oocytes, the translation rate of *mClover3-Ccnb1-3'UTR-A0* is higher in PABPC-depleted oocytes. During the first 6 hours of oocyte maturation, when the poly(A) tail is not long enough, the translation rate is also higher in PABPC-depleted oocytes. By contrast, during the next 6 hours of oocyte maturation, when the poly(A) tail is already extensively elongated, the translation rate is lower in PABPC-depleted oocytes. All these data are consistent with our model. These new data are now described on page 9 line 253-261 of the revised manuscript.

It is worth noting that the regulation of translation during oocyte maturation is much more complicated than in prophase-arrested oocytes. As we have discovered in the previous point, the reporter mRNAs (not mRNA specific or poly(A)-tail-length-specific) are generally translated more efficiently in prophase-arrested oocytes than in maturing oocytes (**Response Fig. 4**). It is therefore possible that the translation efficiency of the same mRNA with the same tail length is also different at different stages of oocyte maturation. Nevertheless, the complexity of the regulation does not affect any of the main conclusions of our manuscript.

Figs. 3-4-5: In Figs. 3-4-5, the authors inject various translation reporters with different lengths of poly(A) tail. The basic assumption of all these experiments is that the reporters will maintain the same length of poly(A) tails in prophase-arrested oocytes during the course of the experiment. This assumption might be incorrect since adenylase/deadenylase activities might be present in the oocytes, even during the prophase arrest, which could act elongating/reducing the length of the poly(A) of the reporters. This has been already suggested in mouse oocytes (10.1093/nar/gkaa010). In this paper, the same translation reporter was produced with different lengths of the poly(A) tail and they were injected in the oocytes arrested in prophase. The rate of translation of these reporters at the beginning was different, because of the different lengths of poly(A) tail, but with time their rate of translation converged to the same level, suggesting that the length of their poly(A) tail has changed. The level of translation to which they were converging depends on the information (or lack of information) present in the 3'UTR.

I would suggest to use a PAT-assay (as the authors have done for endogenous cyclin B1 in Fig. 3B) to measure the length of the poly(A) tails of the reporters (A0, A34, and A>150) to see if it changes after 6h and 12h of injection in oocytes that are maintained arrested in prophase. This should be done for both the mClover3 reporter and the mClover3-Ccnb1 3'UTR reporters, since the information in the 3'UTRs or the lack of UTR could result in a different equilibrium of adenylase/deadenylase activities on the reporter. The stability of the reporter should be measured by qPCR to assess its stability since the absence of poly(A) could result in a partial instability of the reporter.

We thank the reviewer for highlighting this interesting point.

We agree with the reviewer that this paper (10.1093/nar/gkaa010) suggests the presence of adenylase/deadenylase activities in prophase-arrested oocytes, although this paper did not use a PAT assay to confirm this. However, it is unlikely that the potential adenylase/deadenylase activity affects our conclusions for the following reasons. Using the previous **Fig. 4a-c** (new **Fig. 4a-c**) as an example, if the adenylase/deadenylase activity resulted in the gradual conversion of mRNAs with different poly(A) tail lengths (A0, A34, >A150) into mRNAs with the same tail length, making it difficult to see their differences in translation, this would be a problem for us. However, the differences observed between A0, A34 and >A150 in control oocytes were quite large (A0 is 1, A34 is 2.82, >A150 is 9.07), suggesting that the potential adenylase/deadenylase activity was not a problem here. From another point of view, if the presence of adenylase/deadenylase activity gradually made A0, A34 and >A150 mRNAs similar, this means that the actual differences of these mRNAs in control oocytes were even larger than what we observed, which of course does not affect our conclusions. Moreover, in this paper (10.1093/nar/gkaa010), the convergence of translation rates took more than 6 hours, whereas we have already observed a clear phenotype within 3 hours (**new Extended Data Fig. 4a,b** or **previous Extended Data Fig. 4a,b**).

To completely exclude the possibility that the potential adenylase/deadenylase activity affected our experiments, we have now measured the rates of translation under different conditions in the revised manuscript. In control oocytes injected with A0, A34 and >A150 mRNAs, we did not see any convergence of translation rates during the course of the experiment (**new Fig. 4d, blue lines**), suggesting that the potential adenylase/deadenylase activity did not affect our experiments. By contrast, when PABPCs were acutely depleted, the translation rates of these mRNAs became close to each other. Thus, the effect of PABPC Trim-Away is much greater than the potential adenylase/deadenylase activity in prophase-arrested oocytes. These new data are shown in **new Fig. 4d** of the revised manuscript.

Although the potential adenylase/deadenylase activity did not affect our experiments and conclusions, the reviewer's comments reminded us that we need to try to exclude the possibility that acute PABPC depletion changed mRNA poly(A) tail length or stability. To address this point, we unbiasedly selected a few published mRNAs with different poly(A) tail lengths based on some reasonable criteria (**new Extended Data Fig. 4c**), and performed the modified RIBOmap assay to test how acute depletion of PABPCs affects their translation. We found that PABPC depletion increased the translation of short-tailed mRNAs, decreased the translation of long-tailed mRNAs, and had no obvious effect on the translation of *Actb* mRNA, whose tail is intermediate between the other two groups (**new Fig. 4e-i**). These data correlate well with the reporter assay in **new Fig. 4a-d** and further support our conclusions. Particularly, PAT and qPCR assays showed that acute depletion of PABPCs did not alter the poly(A) tail length or stability of these endogenous mRNAs as well as cyclin B1 (**new Extended Data Fig. 4d-i** and **new Extended Data Fig. 5a-c**). These new data are now described on page 8 line 211-222 and page 9 line 237-239 of the revised manuscript.

We would also like to perform the PAT and qPCR assays for all the reporters used in **new Fig. 4a** and **new Fig. 5d** (**previous Fig. 4a,d**). However, performing these experiments is not technically possible. First, two rounds of quantitative injections are required to prepare the oocytes (**new Fig. 4a** and **new Fig. 5d**). It is already very challenging to obtain enough oocytes for the PAT and qPCR assays for just one reporter and under one condition. To prepare enough

oocytes for all reporters under all conditions would be an unimaginable amount of work. Second, to distinguish the reporter mRNAs containing the *Ccnb1* 3'UTR from the endogenous cyclin B1 mRNA in the PAT assay, a forward primer localized to *mClover3* is required. This means that the expected PCR product will be larger than 1 kb (**Response Fig. 2a**). For PAT assays with limited materials, it is necessary to make the PCR products as short as possible for easier amplification and comparison. It is almost impossible to make the PAT assay work if such a long fragment has to be amplified, and the fragment is also too large to reflect a small change in the poly(A) tail length of the reporter mRNA (**Response Fig. 2b**). Third, the PAT assay cannot be applied to mRNAs without any poly(A).

Together, we hope the reviewer agrees that our new data, including the measurement of translation rates (**new Fig. 4d**) and the extensive analyses of endogenous mRNAs (RIBOmap, PAT and qPCR) (**new Fig. 4e-i and new Extended Data Fig. 4c-l**), are sufficient to exclude the possibility that the potential adenylase/deadenylase activity in prophase-arrested oocytes could affect our conclusions.

This control is very important also for Fig. 5 in which the length of poly(A) tail is used to estimate the number of PABPCs bound to the reporter. These experiments are important to validate the model for translation summarized in Fig. 7.

For the experiments in the **previous Fig. 5 (new Fig. 6)**, we have now measured the translation rates of all reporter mRNAs under all conditions in the revised manuscript (**new Extended Data Fig. 6c-e**). Although there were some small fluctuations in translation rates during the course of the experiments, these fluctuations were almost negligible compared to the significant translational repression by DDX6 or LSM14B. Therefore, it is unlikely that the potential adenylase/deadenylase activity could interfere with our experiments. These new data are now described on page 10 line 281-289 of the revised manuscript.

In the revised manuscript, we have now measured the poly(A) tail length of *mClover3-2xPP7-A34* and *mClover3-2xPP7->A150* reporter mRNAs in vitro. The measured poly(A) tail lengths are similar to the expected sizes (**new Extended Data Fig. 6a,b**). The long poly(A) tail in *mClover3-2xPP7->A150* was generated by in vitro polyadenylation, so it is a mixture of different lengths, most of which are longer than 150 nt. We would also like to measure the poly(A) tail length of the reporter mRNAs after they have been injected into oocytes, but this is not technically possible either. In all these experiments, three mRNAs (reporter mRNA, injection control and *tdPCP-Ddx6/Lsm14B*) were co-injected into oocytes and they shared a common sequence coming from the vector at the 3' end of the mRNAs, so we have to choose a forward primer outside this region to distinguish the reporter mRNAs from other mRNAs. We also cannot choose a primer in *2xPP7* or *10xPP7* because these are repeated sequences. The only choice is again *mClover3*, which results in a very challenging PAT assay, especially in the case of *10xPP7*.

We hope the reviewer agrees that the measurement of translation rates (**new Extended Data Fig. 6c-e**) and the measurement of poly(A) tail length in vitro (**new Extended Data Fig. 6a,b**) are sufficient to exclude the possibility that the potential adenylase/deadenylase activity in prophase-arrested oocytes could affect our conclusions.

Fig. 7B: The authors proposed that in the absence of PABPCs, ribosomes are distributed at the same extent on the mRNAs regardless of the length of the poly(A) tail. This has been shown for mClover3 reporter without UTR in Fig. 4A-B-C, and for mClover-CcnB1UTR in Fig. 4E-F-G. However, comparing the same reporter with different poly(A) tails does not allow to evaluate the contribution of the information of the 3'UTR, which also plays a role in determining the level of translation of the reporter, and therefore the ribosome recruitment.

I would suggest the author to compare the translation of mClover-CcnB1UTR and mScarlet-noUTR (both reporters without poly(A) tail) after co-injection in prophase oocytes where PABPCs are acutely depleted. Such an experiment would indicate if the elements present in 3'UTR can still produce a differential recruitment of the ribosomes on the reporter, or if they are not anymore influent when the poly(A) sensing machinery is removed.

We thank the reviewer for this helpful suggestion.

As suggested by the reviewer, we checked how acute depletion of PABPCs affected the translation of *mClover3-Ccnb1 3'UTR-A0* and *mScarlet-A0* (**Response Fig. 5**). PABPC depletion increased the translation of both, although to slightly different extents (**Response Fig. 5**). These data are also consistent with those shown in **new Fig. 4a-d** and **new Fig. 5e-g**.

In our study, we identified two mechanisms that maintain maternal mRNA dormancy in prophase-arrested oocytes. First, a complex of the RNA-binding proteins DDX6, LSM14B and CPEB1 directly represses the translation of cyclin B1 and other CPEB1-bound mRNAs through interacting with their 3'UTR (**new Fig. 2 and new Extended Data Fig. 2**). Second, cytoplasmic poly(A)-binding proteins (PABPCs) indirectly repress the translation of cyclin B1 and other poly(A)-tail-less or short-tailed mRNAs by sequestering the translation machinery on long-tailed mRNAs. It is very likely that these two mechanisms do not interfere with each other, because they act through the 3'UTR and the poly(A) tail, respectively. Thus, for dormant mRNAs such as cyclin B1, even though PABPCs are depleted and the poly(A) sensing machinery is removed, the other translational repression mechanism acting through the 3'UTR is still functioning. This is partially supported by the fact that acute depletion of DDX6 showed a stronger phenotype than acute depletion of PABPCs in terms of increased cyclin B1 translation and premature meiotic resumption (**new Fig. 1f, new Fig. 2f,g and new Fig. 5a-c**). If acute depletion of PABPCs interfered with the effect of DDX6/LSM14B/CPEB1 on cyclin B1 3'UTR, we would expect a stronger phenotype, but this is not the case. On the other hand, although DDX6 Trim-Away showed a stronger phenotype than PABPC Trim-Away, PABPC Trim-Away can further enhance the DDX6 Trim-Away phenotype on both cyclin B1 translation (**previous Fig. 3k-m or new Fig. 3k-m**) and the speed of premature meiotic resumption (**new Extended Data Fig. 5g-i**). All these data support that these two mechanisms act independently. These new data are now described on page 9 line 245-248 of the revised manuscript.

Response Fig. 5: Translation of *mClover3-Ccnb1 3'UTR-A0* and *mScarlet-A0* was both increased upon acute depletion of PABPCs, although the rate of increase was slightly different. Data are shown as mean \pm SD. *mClover3* and *mScarlet* of control IgG were both normalized to “1” to compare their increase in translation upon acute PABPC depletion. *P* values were calculated using two-way ANOVA with Tukey’s *post-hoc* test. Scale bar, 10 μ m.